



# Organic matter dynamics along a salinity gradient in Siberian steppe soils

Norbert Bischoff[1], Robert Mikutta[2], Olga Shibistova[1,3], Reiner Dohrmann[4], Daniel Herdtle[1], Lukas Gerhard[1], Franziska Fritzsche[1], Alexander Puzanov[5], Marina Silanteva[6], Anna Grebennikova[6], Georg Guggenberger[1]

[1]Institute of Soil Science, Leibniz University Hannover, Herrenhäuser Straße 2, 30419 Hannover, Germany
[2]Soil Science and Soil Protection, Martin Luther University Halle-Wittenberg, Von-Seckendorff-Platz 3, 06120 Halle (Saale), Germany
[3]VN Sukachev Institute of Forest, Siberian Branch of the Russian Academy of Sciences, Akademgorodok 50, 660036 Krasnoyarsk, Russian Federation
[4]Federal Institute for Geosciences and Natural Resources, Stilleweg 2, 30655 Hannover, Germany
[5]Institute for Water and Environmental Problems, Siberian Branch of the Russian Academy of Sciences, Molodezhnaya Street 1, 656038 Barnaul, Russian Federation
[6]Faculty of Biology, Altai State University, Prospekt Lenina 61a, 656049 Barnaul, Russian Federation

Correspondence to: Norbert Bischoff (bischoff@ifbk.uni-hannover.de)





**Abstract**

Salt-affected soils will become increasingly important in the next decades as arid and semi-arid ecosystems are predicted to expand as a result of climate change. Nevertheless, little is known about organic matter (OM) dynamics in these soils, though OM is largely controlling soil fertility and represents an important C sink. We aimed at investigating OM dynamics along a salinity and sodicity gradient in soils of the south-western Siberian Kulunda steppe (Kastanozem, Non-sodic Solonchak, Sodic Solonchak) by assessing the organic carbon (OC) stocks, the quantity and quality of particulate and mineral-associated OM in terms of non-cellulosic neutral sugar contents and carbon isotopes ($\delta^{13}$C, $^{14}$C activity), and the microbial community composition based on phospholipid fatty acid (PLFA) patterns. Our hypotheses were that (i) soil OC stocks decrease along the salinity gradient, (ii) the proportion and stability of particulate OM is larger in salt-affected Solonchaks as compared to non-salt-affected Kastanozems, and (iii) sodicity reduces the proportion and stability of mineral-associated OM. Against our first hypothesis, OC stocks increased along the salinity gradient with most pronounced differences between topsoils. In contrast to our second hypothesis, the proportion of particulate OM was unaffected by salinity, thereby accounting for only <10% in all three soil types, while mineral-associated OM contributed to >90%. Isotopic data ($\delta^{13}$C, $^{14}$C activity) and neutral sugars in the OM fractions indicated a comparable degree of OM transformation along the salinity gradient, thus particulate OM was not more persistent under saline conditions. This we attribute to a resilient microbial community composition and function, which was nearly unaffected by salt occurrence, and capable of decomposing OM at a similar rate in salt-affected and non-salt-affected soils. Also our third hypothesis was rejected, as saline-sodic soils contained more than twice as much mineral-bound OC than non-salt-affected soils, what we ascribe to the flocculation of OM and mineral components under higher ionic strength conditions. We conclude that salt-affected soils contribute significantly to the OC storage in the semi-arid soils of the Kulunda steppe while most of the OC is associated to minerals and therefore effectively sequestered in the long-term.





## Introduction

Salt-affected soils occur predominantly in arid and semi-arid environments where rainfall is insufficient to leach salts from the soil (Mavi et al., 2012). They form either anthropogenically as a result of agricultural mismanagement or naturally due to the accumulation of salts from mineral weathering, dust deposition,

precipitation or capillary rise of shallow groundwater tables (Essington, 2004). According to FAO (2001), salt-affected soils include Solonchaks and Solonetzes. While Solonchaks contain high loads of water-soluble salts, Solonetzes are primarily distinguished by $Na^+$ as a dominant cation present on the exchange sites and usually a pH > 8.5. This difference in the composition of salts within both soil types leads to contrasting physico-chemical properties. Solonchaks have a compact soil aggregation, whereas soil particles tend to disperse in Solonetzes,

causing a poor soil structure and the translocation of clay (lessivation) and organic matter (OM) as well as clogging of pores which results in reduced water infiltration, increased surface run-off, and the risk of erosion (Qadir and Schubert, 2002; Sumner, 1993). Salt-affected soils, i.e. both Solonchaks and Solonetzes, respectively, are harsh environments for plants as high salt contents reduce the osmotic potential and subsequently limit plant water uptake (Läuchli and Grattan, 2007). Nutrient uptake is impeded due to ion competition and the high pH,

while the poor soil structure particularly in Solonetzes has adverse effects on soil water balance and plant development (Qadir and Schubert, 2002). As a result, plant residue inputs into the soil are reduced and, thus, lead to small soil OM contents (Wong et al., 2010). However, OM is a key component of soils, being a reservoir for nutrients and determining a soil's agricultural productivity, while, at the same time, it is an important C repository and plays a pivotal role in the course of climate change (Lal, 2004). Particularly by improving soil

structure and increasing the selectivity of exchange sites for $Ca^{2+}$, soil OM can ameliorate sodic soils (Nelson and Oades, 1998; Sumner, 1993).

Independent from soil genesis, salt-affected soils are classified according to their electrical conductivity (EC; in dS m$^{-1}$) and sodium adsorption ratio (SAR) of the saturated paste extract into saline (EC >4 and SAR <13), sodic (EC <4 and SAR >13), and saline-sodic (EC >4 and SAR >13; U.S. Salinity Laboratory Staff, 1954). Both

parameters control the soil structure due to their impact on the dispersion of clay and OM significantly. Numerous studies showed that the desorption of OM from clay particles increases with SAR, while a rise in EC or the proportion of divalent cations counterbalances the dispersing effect of $Na^+$ by inducing flocculation (Mavi et al., 2012; Nelson and Oades, 1998; Setia et al., 2014). High soil pH is likewise supposed to increase losses of organic C (OC) through solubilization of OM (Pathak and Rao, 1998). Peinemann et al. (2005) concluded that in

salt-affected soils mineral-associated OM can be rapidly lost through dispersion and subsequent leaching as dissolved OM, while particulate OM represents a relatively stable fraction as its decomposition is reduced due to an inhibited microbial activity. In line with this, previous work revealed in incubation and field-studies that the microbial decomposition of soil OM is reduced along salinity gradients (Rath and Rousk, 2015; Rietz and Haynes, 2003). However, little is known about microbial functioning in salt-affected soils, and particularly for

Solonchaks and Solonetzes, there are so far no studies available that characterized microbial community compositions.

Though, based on conclusions from sorption-desorption experiments, previous studies noted the sensitivity of mineral-organic associations in salt-affected soils, to date, no study quantified the amount and properties of mineral-associated and particulate OM in these soils. This is surprising, as the occurrence of salt-affected soils is

predicted to increase as a result of climate change due to enhanced aridity (Amini et al., 2016). To date, these soils cover already an area of 831 Mio. ha worldwide (Martinez-Beltran and Manzur, 2005), of which





Solonchaks and Solonetzes constitute about 260 Mio. ha and 135 Mio. ha, respectively (IUSS Working Group WRB, 2014). Thus, our objectives were to (i) elucidate the effect of salinity and sodicity on soil OC stocks, (ii) determine the quantities and properties of functionally different soil OM fractions (particulate *vs.* mineral-associated OM), and (iii) relate our results to changes of the microbial community composition. We approached

this by comparing soil OC stocks, the amount and properties of density-separated OM fractions (contents of hydrolysable non-cellulosic neutral sugars; $\delta^{13}$C and $^{14}$C activity), and the PLFA-based microbial community composition of three soil types representing an increasing impact of salinity and sodicity along a transect in the south-western Siberian Kulunda steppe. Non-cellulosic sugars were chosen as an OM quality parameter, as they enter the soil in large amounts with litter, root residues and plant rhizodeposits as well as by products of

microbial and faunal metabolism and represent a major energy source for heterotrophic soil microbial communities (Cheshire, 1979; Gunina and Kuzyakov, 2015). Additionally, soil aggregate stability was determined to assess the effect of salts, particularly $Na^+$, on the structural stability of the soils. We hypothesized that (i) soil OC stocks decrease along the salinity gradient, because high salinity decreases plant growth and subsequently lowers soil OC inputs, (ii) the proportion and stability of particulate OM is larger in salt-affected

soils as compared to non-salt-affected soils since microbial decomposition and transformation of OM is reduced under high salinity levels, and (iii) sodicity reduces the proportion and stability of mineral-associated OM.

## Material & Methods

### Study site and soil sampling

The study site is located in the south-western Siberian Kulunda steppe which is part of the Altaysky Kray

(Russian Federation). The area belongs to the dry steppe type with a mean annual temperature of 2.6 °C and a mean annual precipitation of 285 mm (climate data from "WorldClim" data base; Hijmans et al., 2005). The studied transect (52°3'36.51"N, 79°36'0.71"E) ranged from a lake over a terraced hillslope to about 5 m upslope the lake (Figure 1). The groundwater table varied from ca. 140 cm next to the lake to >300 cm at the highest point of the transect. Three different soil types developed along the transect primarily as function of the

groundwater table. At shallow groundwater depth close to the lake, Sodic Solonchaks dominated, while Mollic Solonchaks (non-sodic) prevailed backslope with slightly higher groundwater at about 170–180 cm. Upslope the groundwater table reached >300 cm and capillary rise did not reach the soil surface, thus, Haplic Kastanozems and Calcic Kastanozems occurred which were generally grouped as Kastanozems. A detailed soil type classification according to IUSS Working Group WRB (2014) of the analyzed profiles is given in Table S1. We

sampled the soils at plane areas along the terraced slope to avoid the influence of erosion on the soil profiles. Three plots, each with a soil profile down to the groundwater table and locations for plant analyses, were established per soil type; only in the Kastanozems the groundwater was too deep to be reached. Four plots were analyzed on the footslope next to the lake, where site heterogeneity was larger, but one of the four soils was not classified as Sodic Solonchak but as Haplic Solonchak. This soil profile was grouped together with the Mollic

Solonchaks since these soils corresponded to a lower level of sodicity and they were referred to as Non-sodic Solonchaks. Therefore, Kastanozems and Sodic Solonchaks were represented by three soil profiles, while Non-sodic Solonchaks were characterized by four soil profiles. Composite soil samples were taken according to generic horizons in the profiles. Plant samples (shoots and roots) were taken within the plots 5 m distant from around each profile for determination of OC, total nitrogen (TN), and $\delta^{13}$C. The above-ground biomass was





determined in triplicate around each profile by cutting off plants in a 40 cm x 40 cm square and subsequent drying (70°C) and weighing of plant material. The major plant species are listed in Table 1.

**Sample preparation and basic soil analyses**

Samples from generic horizons of the profiles were air-dried and sieved to <2 mm. Visible plant materials were
removed and big clods were gently broken to pass the sieve. An aliquot of the fine earth fraction was dried at 105°C to determine the residual soil water content. Soil bulk density was determined gravimetrically in triplicate for generic horizons by use of a soil sample ring. Soil pH was measured in a 1 : 2.5 (w : v) soil-to-water suspension after equilibration for one day. Carbonate content was analyzed by the Scheibler volumetric method (Schlichting et al., 1995). The texture of the soils was determined according to the standard sieve-pipette method
(DIN ISO 11277, 2002) and the content of oxalate- and dithionite-extractable Fe was analyzed as described in McKeague and Day (1966). Soil aggregate stability was measured based on a method modified from Hartge and Horn (1989) and explained in detail in Bischoff et al. (2016). It was calculated as the difference between the mean weight diameter (MWD) of aggregates of a dry- and a wet-sieving method, expressed as ΔMWD, with a large ΔMWD corresponding to low aggregate stability and a small ΔMWD relating to high aggregate stability.

**Soil salinity parameters**

The content and composition of water-soluble salts was determined by shaking the soil in a 1 : 5 (w : v) soil-to-water suspension at 15 rpm during 1 h and leaving the sample for one day to reach equilibrium. After measuring the EC the extract was centrifuged at 3,000 g for 15 min and filtered through 0.45-μm syringe filters (Cellulose acetate). An aliquot of the extract was measured for $Na^+$, $K^+$, $Ca^{2+}$, and $Mg^{2+}$ with an inductively coupled plasma
optical emission spectrometer (Varian 725-ES; Agilent Technologies, Santa Clara, USA) while another aliquot was analyzed for $Cl^-$, $NO_3^-$, and $SO_4^{2-}$ with an ion chromatograph (ICS-90; Dionex Corp., Sunnyvale, USA). The concentrations of $Na^+$, $Ca^{2+}$, and $Mg^{2+}$ (mmol $l^{-1}$) in the extract were used to calculate the SAR according to Eq. (1).

$$SAR = \frac{Na^+}{(Ca^{2+} + Mg^{2+})^{0.5}}$$
(1)

**Soil mineralogical composition**

X-ray diffractograms of ball-milled <2-mm fractions were recorded with an X'Pert PRO MPD Θ–Θ diffractometer (PANalytical, Almelo, Netherlands) equipped with a Cu anode producing Kα radiation. The powder samples were scanned from 2° to 85° 2Θ with a step size of 0.02° 2Θ and 3 s per step. A subset of samples was evaluated at the micro-scale using a Quanta 600 FEG environmental scanning electron microscope
(ESEM; FEI Company, Hillsboro, USA) with an acceleration voltage of 20 keV. As the analysis was carried out in low-vacuum mode (0.6 mbar), sputtering of the samples with gold or carbon was not necessary. The microscope was equipped with an Apollo XL EDX detector (Ametek Inc., Berwyn, USA).

Clay mineralogical analyses were carried out for one representative soil profile of each soil type. Clay fractions (<2 μm) were obtained by pre-treating the soil with acetic acid (removal of carbonates), $H_2O_2$ (removal of OM),
and dithionite-citrate (removal of iron oxides), subsequent separation by sedimentation (Stoke's law) and final $Mg^{2+}$ saturation to cause flocculation and thus easier handling of samples. X-ray diffraction patterns were recorded using the same system and settings as for the powder analyses of bulk soil but with Co-Kα radiation





generated at 40 kV and 40 mA. Oriented mounts were prepared on porous ceramic tiles to avoid segregation of fine particles during sedimentation (Dohrmann et al., 2009) and scanned from 2° to 35° 2Θ with a step size of 0.02° 2Θ and 4 s per step. Sample quantity allowed only for two treatments for qualitative analysis: (i) $Mg^{2+}$, (ii) $Mg^{2+}$ + ethylene glycol. The ethylene glycol treatment was used as it detects expandable clay minerals like

smectite, which strongly affect the physical properties of sodic soils.

**Determination of organic carbon, δ$^{13}$C, and total nitrogen**

Ball-milled <2-mm fractions were measured for OC and TN as well as for δ$^{13}$C via dry combustion in an Elementar vario MICRO cube C/N Analyzer (Elementar Analysensysteme GmbH, Hanau, Germany) coupled to an IsoPrime IRMS (IsoPrime Ltd, Cheadle Hulme, UK) after removing inorganic C by fumigation with HCl and

subsequent neutralization over NaOH pellets (modified from Walthert et al., 2010). The measured δ$^{13}$C values were corrected by calculating response factors from standard compounds ($CaCO_3$, cellulose, caffein) and expressed in the delta notation related to the Vienna Peedee-Belemnite-Standard (0‰). The complete removal of inorganic C from all samples was confirmed by δ$^{13}$C values which are in the typical range of soil OM (-22.5‰ to -28.1‰).

**Density fractionation and $^{14}$C analysis**

Density fractionation (modified after Golchin et al., 1994) separated the soil into a light fraction (LF), containing mostly particulate OM, and a heavy fraction (HF), consisting of mineral-associated OM as well as mineral components free of OM. As particulate OM contents are mostly very low in the subsoil, we fractionated the soil only until the first C horizon of each profile. In brief, 25g soil was weighted in duplicate into beakers and 125ml

sodium polytungstate (SPT; ρ = 1.6 g cm$^{-3}$) was added, gently stirred with a glass rod and ultra sonification was applied with an energy input of 60 J ml$^{-1}$during 8 min to break down aggregates. After centrifugation at 3,000 g for 20 min the LF was separated from the HF by decanting the floating LF on polyethersulfone filters and repeating the procedure if the separation between both fractions was insufficient. LF remaining on the filter was washed with deionized water to remove residual SPT until the washing solution had an EC <60 µS cm$^{-1}$. The HF

remaining in the beaker was washed with deionized water until the EC of the washing solution was <100 µS cm$^{-1}$, but at maximum four times in the salt-affected soils, as no residual SPT was detected afterwards by ESEM–EDX analysis, which was carried out with a Quanta 200 FEG environmental scanning electron microscope (FEI Company, Hillsboro, USA) coupled to an XL–30 EDX detector (Ametek Inc, Berwyn, USA). The washing solutions of both LF and HF, respectively, were collected, filtered through 0.45-µm syringe filters (PVDF), and

measured for non-purgeable OC with a LiquiTOC (Elementar Analysensysteme GmbH, Hanau, Germany) to account for the loss of OC during washing of the samples (mobilized OC, MobC; Gentsch et al., 2015). The LF and HF were freeze-dried, weighted, homogenized in a mortar, and subsequently measured for OC and TN as well as δ$^{13}$C as described in Sect. 2.5, after removal of inorganic C. The mobilized OC was added to the OC content of the LF or HF, respectively.

Three representative soil profiles were selected, one per soil type, for analysis of $^{14}$C activities of OM fractions at the Max Planck Institute for Biogeochemistry Jena (Germany). Inorganic C was removed by 2M HCl until pH remained <3.5 and samples were subsequently neutralized with 2M NaOH to pH 6. After freeze-drying $^{14}$C analysis was performed with a 3MV Tandetron$^{TM}$ AMS $^{14}$C system (Steinhof et al., 2011) and $^{14}$C isotope



activities were converted to percent modern carbon (pMC) according to Steinhof (2013), while pMC was defined according to Stuiver and Polach (1977), see Eq. (2):

$$pMC = \frac{A_{SN}}{A_{abs}} \times 100\% \qquad (2)$$

where $A_{SN}$ is the normalized sample activity and $A_{abs}$ corresponds to the activity of the absolute international
standard; both activities were background-corrected and $\delta^{13}$C-normalized. OxCal 4.2 software (University of Oxford) was used to calculate conventional $^{14}$C ages by selecting the IntCal13 calibration curve (Reimer et al., 2013), if pMC was <100%, and the calibration curve from Hua et al. (2013), if pMC was >100%.

**Biomarker analyses**

**Non-cellulosic neutral sugars**

Non-cellulosic neutral sugars were analyzed in the LF and HF from generic horizons of each soil profile. In the LF neutral sugars were only analyzed in some of the topmost horizons, as its content was too low in most samples to provide sufficient material. Additionally, neutral sugars were determined in plant material (shoots and roots). Neutral sugars were analyzed slightly modified according to Rumpel and Dignac (2006), including the EDTA purification step from Eder et al. (2010). In brief, 600mg of HF and 50mg of LF or plant material was
hydrolyzed in 4M trifluoroacetic acid (TFA) at 105°C during 4 h after 1.5ml myo-inositol was added as an internal standard. After cooling to room temperature the extract was filtered through glassfiber filters (Whatman™ GF6) and TFA was removed in a rotary evaporator. The samples were redissolved in ultrapure water and the pH was adjusted to 4–5 by adding $NH_3$. Ferric Fe was complexed by adding 4ml EDTA and incubating the samples in the dark during 10min. From now on darkened glassware was used to prevent
photolysis of Fe(III) ligand complexes. After freeze-drying and adding two drops of $NH_3$ the reduction of aldoses to their corresponding alditols (derivatization) was performed at 40°C during 1.5 h with $NaBH_4$ dissolved in dimethyl sulfoxide. Acetylation was carried out by adding 2ml acetic anhydride and 0.2ml glacial acetic acid, thereby using methylimidazole as a catalyst. Ice-cold deionised water was added after 10 min to stop the reaction. Sugar monomers were extracted by liquid-liquid extraction with dichloromethane and subsequently
measured by gas chromatography on a 7890A GC system (Agilent Technologies, Santa Clara, USA) equipped with a SGE forte GC capillary column (0.25mm diameter and 0.25µm film thickness; SGE Analytical Science, Melbourne, Australia) and a flame ionization detector, using He as a carrier gas. External standards were used to detect eight different sugars: arabinose, xylose and ribose (pentoses), galactose, glucose and mannose (hexoses), and fucose and rhamnose (desoxysugars).

**Phospholipid fatty acids**

Directly after sampling, sieving to <2 mm and removing visible plant materials, 1.0–1.5g field-moist soil was weighted into cryovials and 3ml RNAlater® was added to prevent sample degradation (Schnecker et al., 2012). An aliquot was dried at 105°C to determine the soil water content. The cryovials were kept cool until they were frozen to –20°C within 72 h. For PLFA analysis we used a modified method from Gunina et al. (2014). Briefly,
samples were transferred from cryovials into test tubes and washed with ultrapure water to remove residual RNAlater®. Lipids were extracted twice with a chloroform-methanol-citrate buffer (1:2:0.8 v/v/v) and separated into glycolipids, neutral lipids, and phospholipids by solid phase extraction with activated Silica gel (Sigma Aldrich, pore size 60Å, 70–230 mesh). Phospholipids were derivatized into fatty acid methyl esters (FAME)




with 0.5M NaOH dissolved in methanol and with BF$_3$ as catalyst. FAME were analyzed with a 7890A GC system (Agilent Technologies, Santa Clara, USA) equipped with a 60m Zebron ZB-5MSi capillary GC column (0.25mm diameter and 0.25μm film thickness; Phenomenex, Torrance, USA) and a flame ionization detector, using He as a carrier gas. As an internal standard we used nonadecanoic acid (FA 19:0) and 17 fatty acids were

used as external standards. Peak identification of the internal standard turned out as problematic in the salt-affected topsoils. Therefore we could not reliably quantify individual PLFA but only their relative proportion in the sample. As a result the sum of all PLFA was not used as a proxy of the microbial biomass contents but PLFA were used to characterize the composition of functional microbial groups. We applied a principal components analysis (PCA) on the relative distribution of all 17 PLFA to identify clusters of correlated PLFA, which

presumably derive from identical microbial functional groups. The assignment of individual PLFA to certain microbial groups based on the PCA was in agreement with the literature (Frostegård et al., 2011; Olsson, 1999; Ruess and Chamberlain, 2010; Zelles, 1999). Thus, the following PLFA were used to distinguish functional microbial groups: 18:2ω6,9 and 18:1ω9c as marker for saprophytic fungi (SAP), 16:1ω5c to identify arbuscular mycorrhizal fungi (AMF), i15:0, a15:0, i16:0, i17:0 and a17:0 were related to gram-positive bacteria (Gram+),

10Me16:0 characterized actinomycetes (Actino), 16:1ω7c and 18:1ω7c identified gram-negative bacteria (Gram–), and 14:0, 15:0, 17:0 and 18:0 related to unspecific bacteria (UnspBact). The PLFA Cy19:0 and 20:4ω6c were not used as markers for microbial groups as they hardly reached the detection limit and were sometimes difficult to distinguish from other unspecific peaks in the gas chromatogram.

**Calculation of organic carbon stocks**

Organic C stocks (Mg ha$^{-1}$) were calculated according to Poeplau & Don (2013) for all horizons and the entire soil profile as well as until 1m depth using Eq. (3):

$$OCstock = \sum_{i=1}^{n} \frac{FSM_i}{V_i} \times C_i \times D_i \tag{3}$$

where $n$ is the number of horizons, $FSM$ is the fine-earth soil mass (g), $V$ is the volume (cm$^3$), C is the OC content (% of soil mass) and $D$ is the length of the horizon (cm).

**Statistical analyses**

Data analysis was performed in R, version 3.2.5 (R Core Team, 2016). From replicated measurements we calculated arithmetic means and standard errors. To test for the effect of soil type on above-ground plant biomass a linear mixed effects model was fitted (package lme4; Bates et al., 2012). We accounted for the nested structure of sampling, i.e. the soil type was used as fixed effect while the soil profiles (of each soil type) were included as

random effects. Residuals and random effect estimates of the fitted model were visually assessed by Q-Q-normal plots but no deviations from normality were observed. The difference of the response variable between the soil types was tested based on the linear mixed effects model fit, including corrections for multiple comparisons (analogous to the Tukey test), with Satterthwaite degrees of freedom, on the basis of the R packages lsmeans (Lenth and Herve, 2015), lmerTest (Kuznetsova et al., 2015), and multcomp (Hothorn et al., 2008). Soil sample

related parameters were analyzed descriptively, as their sample size was only 3–4 per soil type, which was insufficient for statistical hypothesis testing. Analysis of data from PLFA and neutral sugars involved the consideration of several response variables which was done by PCA, thereby adding confidence regions (68%) for the group centroids of the analyzed factor variables. Figure 1 was drawn in Inkscape, while the other graphs were generated using ggplot2 (Wickham, 2009).



## Results

### Basic soil and site properties

The soil moisture during sampling (% of dry weight) was very small in the Kastanozems (3.6–4.5%) and larger in the salt-affected soils with shallow groundwater table (Non-sodic Solonchaks: 14.9–20.5%, Sodic Solonchaks:
16.4–30.6%; Table 2). The pH in the Kastanozems increased from about 7 in the topsoil to 9 in the subsoil, while the Solonchaks revealed a nearly constant pH throughout the soil profile between 8.5 and 9. While Kastanozems had no carbonates in the topsoil, the carbonate content peaked in the Ck horizon with $51 \pm 12$ mg g$^{-1}$ (Table 2). The salt-affected soils exhibited larger carbonate contents, between $53 \pm 16$ mg g$^{-1}$ and $152 \pm 34$ mg g$^{-1}$ in the Non-sodic Solonchaks and $115 \pm 49$ mg g$^{-1}$ and $264 \pm 22$ mg g$^{-1}$ in the Sodic Solonchaks. The aggregate stability
was larger in Kastanozems and Sodic Solonchaks ($\Delta$MWD: $0.41 \pm 0.06$ mm and $0.33 \pm 0.03$ mm, respectively) than in Non-sodic Solonchaks ($1.02 \pm 0.29$ mm; Table 2). The Kastanozems consisted mostly of sandy loam, while the Solonchaks were more loamy with larger clay and silt contents. Oxalate- and dithionite-extractable Fe was consistently low in all three soil types (<0.4 mg g$^{-1}$Fe$_O$, <5 mg g$^{-1}$Fe$_D$; Table 2).

### Soil salinity parameters

The EC$_{1:5}$ was small (<250 $\mu$S cm$^{-1}$) in the Kastanozems with a slight increase from top- to subsoil, while the largest EC$_{1:5}$ in the Solonchaks was found in the topsoil (Table 2). In the Non-sodic Solonchaks the EC$_{1:5}$ decreased from $3416 \pm 1053$ $\mu$S cm$^{-1}$ in the topsoil to $796 \pm 333$ $\mu$S cm$^{-1}$ in the subsoil, while the Sodic Solonchaks had the largest EC$_{1:5}$ with $5350 \pm 1476$ $\mu$S cm$^{-1}$ in the topsoil and the smallest EC$_{1:5}$ with $1093 \pm 702$ $\mu$S cm$^{-1}$ in the subsoil. The SAR$_{1:5}$ revealed a similar pattern, with small SAR$_{1:5}$ (<2) in the Kastanozems and
larger values in the Solonchaks (Table 2). In the Non-sodic Solonchaks the SAR$_{1:5}$ dropped from $9.6 \pm 2.2$ in the topsoil to $3.9 \pm 1.0$ in the subsoil, while Sodic Solonchaks had the largest SAR$_{1:5}$ with $36.0 \pm 10.4$ in the topsoil and $8.0 \pm 4.6$ in the subsoil. The composition of water-soluble anions and cations was different in the two salt-affected soils (Figure S1). While the Non-sodic Solonchaks had an almost balanced concentration of SO$_4^{2-}$ and Cl$^-$ on the one hand, and Na$^+$, Ca$^{2+}$ and Mg$^{2+}$ on the other hand, the Sodic Solonchaks were dominated by SO$_4^{2-}$
and Na$^+$, with smaller quantities of Cl$^-$.

### Soil mineralogical composition

The three soil types had a quite homogenous mineralogical composition, dominated by quartz and feldspars as well as calcite and dolomite in the carbonate-rich horizons, whereas almost all samples showed small quantities of amphibole and muscovite (Figure S2). In the Solonchaks also gypsum was present. Calcite and dolomite XRD
peaks were very broad, peak broadening is related to very fine crystallite sizes. The clay fraction showed small amounts of illite, kaolinite, and chlorite, while smectites were partially present in the subsoil, and in the Sodic Solonchak also in the topsoil. In the smectite-rich horizons, the quantities of smectite and illite exceeded those of chlorite and kaolinite significantly (Figure S3). In the Solonchaks, the quantities of water-soluble salts were small when related to the bulk soil. Mass balance calculations (data not shown) and analyses by ESEM–EDX
(Figure S4) revealed that water-soluble salts mostly consisted of thenardite ($\alpha$-Na$_2$SO$_4$) and halite (NaCl), but also bischofite (MgCl$_2 \cdot$ 6H$_2$O) could be present.




### Soil organic carbon stocks

Soil OC stocks increased with salinity and sodicity from Kastanozems over Non-sodic Solonchaks to Sodic Solonchaks (Figure 2). Differences were most pronounced in the topsoils, while subsoil OC stocks were similar between the soil types. Down to a depth of 100 cm Kastanozems had 70.9 ± 2.8 Mg OC ha$^{-1}$, Non-sodic

Solonchaks 94.2 ± 6.9 Mg OC ha$^{-1}$ and Sodic Solonchaks 129.5 ± 25.6 Mg OC ha$^{-1}$. Thus, OC stocks in Non-sodic Solonchaks were 32.8 ± 9.7% larger than in Kastanozems and OC stocks of Sodic Solonchaks exceeded those of Kastanozems even by 82.6 ± 36.1%. The C : N ratios were comparable along the salinity gradient and ranged from about 10 in the topsoil to 5–8 in the subsoil (Table S2).

### Soil organic matter fractions

**Organic carbon contents and isotopic composition**

All three soil types were dominated by HF–OC with >90% of bulk OC, while LF–OC accounted for <10% (Table 3). The proportion of HF–OC revealed no clear depth gradient within the soil profiles. The OC content of the HF increased in A horizons with salinity and sodicity from Kastanozems (7.7 ± 0.3 mg g$^{-1}$) to Non-sodic Solonchaks (18.3 ± 2.7 mg g$^{-1}$) to Sodic Solonchaks (19.3 ± 5.0 mg g$^{-1}$), while OC contents were similar in the

subsoil (Table 3). OC contents in the LF were smaller in the Kastanozems (120–219 mg OC g$^{-1}$) than in Non-sodic Solonchaks (197–279 mg OC g$^{-1}$) and Sodic Solonchaks (247–265 mg OC g$^{-1}$; Table 3). Kastanozems and Non-sodic Solonchaks had the largest LF-OC contents in the subsoil but LF-OC contents were equal over depth in the Sodic Solonchaks. HF material was enriched in δ$^{13}$C as compared to LF material (Table 3). In the LF δ$^{13}$C ratios ranged from -27.5‰ to -26.4‰ (Kastanozems), -27.0‰ to -28.1‰ (Non-sodic Solonchaks) and -24.3‰ to

-26.9‰ (Sodic Solonchaks). Remarkably, the δ$^{13}$C ratios in the LF decreased from top- to subsoil in the Solonchaks, while the Kastanozems revealed a typical increase of δ$^{13}$C ratios from top- to subsoil. The δ$^{13}$C ratios of the LF were similar to the root signals of the plants, while no relation to the shoot signals was apparent (Figure 4). Ratios of δ$^{13}$C in the HF were comparable between the three soil types and ranged from -23.8‰ to -23.0‰ in the Kastanozems, from -23.3‰ to -22.8‰ in the Non-sodic Solonchaks and from -23.4‰ to -22.5‰

in the Sodic Solonchaks (Table 3). As residual SPT had to be removed during density fractionation for subsequent determination of OC parameters, all samples were washed with deionized water (see Sect. 2.6). This resulted in a loss of HF material. About 8–29 mg HF g$^{-1}$ soil was lost in Kastanozems, while the loss was larger in salt-affected soils due to the high solubility of salts and accounted for 61–86 mg HF g$^{-1}$ soil in Non-sodic Solonchaks and 46–76 mg HF g$^{-1}$ soil in Sodic Solonchaks, with larger losses in samples with high EC (Table 3).

Despite larger HF losses were observed in Solonchaks, the percentage of MobC related to bulk OC was small in these soils (maximally 9.4 ± 1.6%), while Kastanozems had larger proportions of MobC (15.6 ± 0.5% to 45.7 ± 12.0%). This indicates that the water-soluble salts in the salt-affected soils were mostly not associated with OC. The quantities of MobC from the LF were larger in salt-affected soils and accounted for up to 258 mg OC g$^{-1}$ LF, but maximally 3.4% of bulk OC in all three soil types (Table 3). The proportion of MobC increased with depth

in both LF and HF, respectively. The $^{14}$C activities in the LF were similar in the Kastanozem and the Sodic Solonchak and amounted mostly >100 pMC (Table 3), corresponding to recent C with $^{14}$C ages of maximally 60 years B.P. In the Non-sodic Solonchak the $^{14}$C activity was >100 pMC in the topmost horizon (Az1) but lower in the underlying horizons, i.e. 91.67 pMC (ca. 730 years B.P.) in the Az2 horizon and 93.86 pMC (ca. 580 years B.P.) in the Bkz horizon, respectively. This untypically high age of LF material indicated a possible





contamination with HF material. The [14]C activities in the HF were smaller than in the LF, corresponding to higher [14]C ages, and no trend related to the three soil types was apparent. Remarkably, [14]C activities increased from ca. 30 cm depth to 50–60 cm depth after a typical decrease from the topsoil. The [14]C activities in the HF corresponded to [14]C ages of 150–950 years B.P. in the topsoil horizons and 1200–2900 years B.P. in the

underlying horizons, while the highest [14]C age occurred in the comparatively deep Cz horizon (ca. 90 cm) of the Non-sodic Solonchak with 4600 years B.P.

**Non-cellulosic neutral sugars**

The neutral sugar content of the LF from the topmost horizons was similar in the Kastanozems and the Non-sodic Solonchaks with $47 \pm 5$ mg g$^{-1}$ and 46 mg g$^{-1}$, respectively, while Sodic Solonchaks contained more neutral

sugars ($105 \pm 27$ mg g$^{-1}$; Table 3). Related to the OC content, sugar contents were comparable between all soil types and ranged from 328–410 mg g$^{-1}$ OC. The HF contained less sugars than the LF, thereby sugar contents decreased from top- to subsoil according to the decrease of OC contents (Table 3). In topsoils sugar contents of the HF increased from Kastanozems ($1.0 \pm 0.2$ mg g$^{-1}$) over Non-sodic Solonchaks ($3.1 \pm 0.6$ mg g$^{-1}$) to Sodic Solonchaks ($5.7 \pm 0.8$ mg g-1), while sugar contents were similar in the subsoil. Based on the OC content, sugar

contents were similar in the Kastanozems and Non-sodic Solonchaks and ranged between 136–172 mg g$^{-1}$ OC, with no clear depth gradient. Sodic Solonchaks contained more sugar per g OC than the other two soil types, with $322 \pm 61$ mg g$^{-1}$ OC in the topsoil and smaller sugar contents in the subsoil (165 mg sugar g$^{-1}$ OC). The averaged proportion of each sugar in the total sugars was as following: xylose ($27 \pm 8\%$), glucose ($20 \pm 2\%$), arabinose ($19 \pm 2\%$), galactose ($18 \pm 3\%$), mannose ($7 \pm 3\%$), rhamnose ($5 \pm 1\%$), fucose ($3 \pm 1\%$), and ribose ($1$

$\pm 1\%$; data not shown).

The PCA of neutral sugars from plants, LF and HF material revealed two significant components (eigenvalue > 1), the first component (PC1) with 54.9% explained variance and the second component (PC2) related to 18.7% explained variance (Figure 5). The composition of neutral sugars was different between plants, LF material and HF material, while differences between the three soil types were smaller. Plants of all soil types were enriched in

xylose and those of salt-affected soils also in arabinose, while HF material of all soils was augmented with mannose, galactose, fucose, ribose, and rhamnose. Differences between soil types were apparent with respect to arabinose and glucose. In the Kastanozems OM in the LF and HF became enriched in arabinose during decomposition of plant material, while the opposite was observed in the salt-affected soils (see also Figure S5). The relative proportion of glucose remained similar in the Kastanozems but increased in the salt-affected soils in

the course of decomposition (see also Figure S6). However, on the whole, neutral sugars in LF but also HF material were similarly altered in all three soil types with respect to their initial composition in the plant tissue, as indicated by a comparable shift of the three fractions in all soil types along the first axis in the biplot, suggesting a comparable degree of soil OM alteration between the soil types.

**Phospholipid fatty acids**

The relative contribution of PLFA observed within the PLFA profiles was as follows: PLFA from unspecific bacteria ($36.7 \pm 2.2\%$), Gram+ ($25.6 \pm 0.7\%$), Gram– ($11.9 \pm 1.3\%$), SAP ($11.3 \pm 0.9\%$), AMF ($8.4 \pm 1.8\%$) and from actinomycetes ($6.1 \pm 0.6\%$). Thus, bacterial PLFA constituted $80.4 \pm 1.1\%$ while fungal PLFA represented $19.6 \pm 1.1\%$ of the analyzed fatty acids. The PCA of the PLFA-based microbial groups extracted two significant components (eigenvalue >1) and showed a clear differentiation between bacterial and fungal PLFA (Figure 6),





the former stretching along the first component (PC1) and the latter correlating with the second component (PC2). Accordingly, bacterial PLFA explained 57.8% of the variability of total PLFA, while fungal PLFA corresponded to 22.0% of the total variability. PLFA of Gram+, Gram– and actinomycetes were positively correlated with each other, but had a negative correlation to the group of unspecific PLFA. Among the fungal
PLFA, those of AMF correlated negatively to those of SAP. Differences in the microbial community composition existed between soil horizons and were largely explained by the variability of bacterial PLFA, with a larger abundance of Gram+, Gram– and actinomycetes in topsoil horizons and a larger abundance of unspecific PLFA in the subsoil (Figure 6). Changes of the microbial community composition between the three soil types were less pronounced and mostly due to a larger variability of fungal PLFA in the Solonchaks as compared to
the Kastanozems, whereas the composition of bacterial PLFA was similar between all soils. However, the fungi : bacteria ratio was rather constant between the three soil types and amounted to $0.22 \pm 0.03$ in Kastanozems, $0.28 \pm 0.05$ in Non-sodic Solonchaks, and $0.27 \pm 0.03$ in Sodic Solonchaks, with slightly larger fungi : bacteria ratios in the subsoils of the Solonchaks (data not shown).

**Discussion**

**Soil OC stocks along the salinity gradient**

Salt-affected soils, such as Solonchaks, are normally characterized by poor plant growth resulting in small soil OC inputs and subsequently low soil OC stocks (Wong et al., 2010). Muñoz-Rojas et al. (2012), for example, reported soil OC stocks in Solonchaks of southern Spain in 0–75cm depth of 53.6 Mg ha$^{-1}$ (coefficient of variation (CV): 60%) under shrub and/or herbaceous vegetation. Batjes (1996) calculated in the framework of a
global meta-analysis average soil OC stocks of Solonchaks of 42 Mg ha$^{-1}$(CV: 67%) in 0–100 cm depth, while he noted that particularly Mollic Solonchaks had much larger soil OC stocks of 101 Mg ha$^{-1}$ (CV: 44%). Kastanozems, on the other hand, contained on average 96 Mg ha$^{-1}$ (CV: 50%) in the first meter, at which Haplic Kastanozems had soil OC stocks above that average of 138 Mg ha$^{-1}$ (CV: 44%; Batjes, 1996). Based on data from Bischoff et al. (2016), we calculated soil OC stocks in Kastanozems of the dry steppe type of the Kulunda
steppe down to 60 cm, which accounted for $110 \pm 6$ Mg ha$^{-1}$. All of the previously published data confirm that salt-affected soils like Solonchaks have normally smaller OC stocks than the non-salt-affected Kastanozems. Contrary, in our study, salt-affected soils had larger OC stocks as compared to the nearby Kastanozems. With average OC stocks of $70.9 \pm 2.8$ Mg OC ha$^{-1}$ in 0–100 cm depth of the Kastanozems, the values were clearly below those observed by Batjes (1996) and calculated from Bischoff et al. (2016). On the other hand, average
OC stocks of $94.2 \pm 6.9$ Mg OC ha$^{-1}$ and $129.5 \pm 25.6$ Mg OC ha$^{-1}$ in 0–100 cm of the Non-sodic Solonchaks and Sodic Solonchaks, respectively, were clearly above the values reported by Batjes (1996) and Muñoz-Rojas et al. (2012). Larger OC stocks in salt-affected soils than in Kastanozems are also in contrast to earlier work which found a negative effect of salinity on soil OC stocks (reviewed by Wong et al., 2010). Possible reasons for the observed differences are climatic variations between the studies (strong aridity in the Spanish Solonchaks from
Muñoz-Rojas et al., 2012) or alterations in soil texture (finer textured Kastanozems in the study from Bischoff et al., 2016) which may eventually change the soil water balance and thus plant growth and soil OC inputs. Moreover, during sampling we observed very dry conditions in the Kastanozems (only $4 \pm 0.3\%$ soil water related to dry soil mass), while the Solonchaks were generally wetter due to their shallow groundwater table (15–30% soil water, Table 2), such that in these soils water stress was mostly related to a small osmotic potential.



Overall, the water stress in the three soil types could have been similar, either as a result of matric stress or osmotic stress, leading to comparable moisture conditions for plant growth. Accordingly and in contrast to previous work, along our transect plant growth (as measured by above-ground biomass) was not reduced under high salinity (Table 1). As this was discussed as a prerequisite of reduced OC stocks at elevated salinity (Wong

et al., 2010), it can be one reason why our study revealed different results. As the $\delta^{13}$C ratios suggested that soil OM was mostly root-derived in the studied soils (Figure 4), one might argue that above-ground biomass is a poor proxy for soil OC input. However, under the assumption that root residue inputs are correlated with the above-ground biomass (evidence is given by Titlyanova et al. (1999) who observed significant correlations (p < 0.01, R > 0.5) between the above-ground and below-ground biomass of typical plants in Siberian grasslands), we

might conclude that both, above-ground and below-ground soil OC inputs, were comparable between all three soil types.

Moreover, Wong et al. (2010) argued that small OC stocks in salt-affected soils can also be the result of erosion-induced OC losses, as particularly sodic soils are prone to erosion. Since we paid particular attention to the fact that all soils were not affected by erosion, we can rule out erosion as a factor that modified OC stocks in our

study. Summing up, our first hypothesis has to be rejected since soil OC stocks have not decreased along the salinity gradient in contrast to previous observations from comparable soils (Figure 2). As the quantity of plant biomass was not reduced under high salinity, we consider this as main reason for the large OC stocks in salt-affected soils.

**Partitioning and composition of soil OM in different functional OM fractions**

Considering processes of soil OC stabilization, semi-arid soils should have large proportions of particulate OC, as the formation of stable mineral-organic associations is attenuated due to low water availability and a high soil pH (Kleber et al., 2015). However, in the semi-arid soils of the studied transect particulate OC contributed <10% of bulk OC, while mineral-bound OC accounted for >90% (Table 3). This contrasts observations from steppe soils (mostly Chernozems) of European Russia (Breulmann et al., 2014; Kalinina et al., 2011), Canada (Plante et

al., 2010), or China (Steffens et al., 2010), where particulate OC represented >20% of bulk OC. Nevertheless, our results are in line with Bischoff et al. (2016) who reported that maximally 10% of OC was present in particulate OM in Chernozems and Kastanozems of the Kulunda steppe. Thus, we support previous observations from this region and conclude that mineral-bound OM is the dominant OM fraction in both, salt- and non-salt-affected soils of the studied region.

Against our second hypothesis, salt-affected soils contained similar proportions of particulate OC like non-salt-affected soils, with 4–8% particulate OC in all three soil types (Table 3). Comparable $^{14}$C activities in the LF of the three soil types (small $^{14}$C activities in the Non-sodic Solonchak were probably due to a contamination with HF material) indicated a similar turnover of particulate OM, thus contradicting our hypothesis of increased stabilization of particulate OM under high salinity levels. Peinemann et al. (2005) concluded, based on OC

determinations in particle-size separates and analyses of lignin components along a salinity gradient in the Argentinian Pampa, that particulate OM is a relatively stable fraction in salt-affected soils due to a reduced microbial transformation of the plant-derived residue inputs. This is not corroborated by our results. The isotopic C composition ($^{14}$C activity, $\delta^{13}$C) and the composition of neutral sugars suggest a comparable alteration of OM (i.e. degree of OM decomposition) between the three soil types (Figure 3-5). Furthermore, Peinemann et al.

(2005) supposed that mineral-bound OM is relatively susceptible to losses in salt-affected soils due to weak





chemical bonding and subsequently weak OM stabilization. Against our third hypothesis, the OC content of the HF of the salt-affected soils was more than twice as large as of the non-salt-affected Kastanozems (Table 3). Moreover, during washing of the density-separates (SPT removal) relatively less OC was mobilized from the HF of the salt-affected soils (3–10% MobC) than from the HF of the Kastanozems (16–46% MobC, Table 3),

suggesting a lower chemical stabilization of mineral-bound OM in the non-salt-affected soils.

We explain the large contents of mineral-associated OC under high salinity levels by consideration of basic chemical principles. According to Sumner (1993), dispersion of clay minerals is only possible below their critical flocculation concentration (CFC). This concept relates the dispersive effect of $Na^+$ on the soil structure to the corresponding salt concentration of the soil solution (Rengasamy et al. 1984; Sumner et al. 1998). The

authors classified soils into *flocculated*, *potentially dispersive* and *dispersive* depending on the EC and SAR of the soil water extract. Sumner et al. (1998) classified soils with large proportions of non-swelling illitic clays, while Rengasamy et al. (1984) considered soils with swelling 2:1 clays, similar to the smectite-rich soils of our study. According to their classification, all of the salt-affected soils of our study fall into the category *flocculated*; even A horizons of the Sodic Solonchaks with an average SAR of $36 \pm 10$ remain flocculated,

presumably due to the high electrolyte concentration as indicated by a large EC of $5350 \pm 1476$ μS cm$^{-1}$ (Table 2). This is underpinned by the large aggregate stability of the Sodic Solonchaks (Table 2) and the lack of clay lessivation or OM translocation, processes for which the dispersion of clays and OM is one prerequisite. Similarly, Setia et al. (2013, 2014) confirmed that the dispersive effect of $Na^+$ on OM and mineral components is only evident at low electrolyte concentrations, particularly at low concentrations of divalent cations like $Ca^{2+}$.

Nelson and Oades (1998) showed that the solubility of $Na^+$–coated OM is larger than that of OM coated with $Ca^{2+}$. Thus, particularly in the Non-sodic Solonchaks where $Ca^{2+}$ is a dominant cation in the soil solution (Figure S1), the solubility of OM can be reduced. In summary, mineral-bound OM is stabilized in the studied salt-affected soils as the large electrolyte concentration in the soil solution promotes the flocculation of OM and minerals. On the other hand, particulate OM is not as stable in salt-affected soils as previously assumed, as the

degree of decomposition of this OM fraction was similar between salt-affected and non-salt-affected soils.

**Microbial community composition along the salinity gradient**

Microbial communities are sensitive to environmental changes and react to differences in the osmotic and matric potential (Rath and Rousk, 2015; Schimel et al., 2007). Particularly fungi but also Gram+ are thought to be more resistant against drought than Gram− due to their ability to produce osmolytes (Schimel et al., 2007). However,

previous work on differences of the microbial community composition along salinity gradients could not support the view that fungi are superior to bacteria under water stress, i.e. high salinity, as several studies observed even a negative relationship between fungal abundance and salinity (Baumann and Marschner, 2011; Chowdhury et al., 2011; Pankhurst et al., 2001). This suggests that in salt-affected soils not only drought dictates the abundance of certain microbial groups but that also toxic effects of certain ions or impeded nutrient uptake as a result of ion

competition may exist. In our study, the fungi : bacteria ratio was not related to the salinity gradient and was similar between the three soil types. Moreover, the fungal PLFA composition revealed a larger variability in the salt-affected soils, indicating that these soils have a more variable fungal community composition. Both results can have consequences, if we consider that fungi are thought to be the primary decomposers of particulate OM (evidence is given by Bossuyt et al., 2001; Frey et al., 2000; Six et al., 2006): a fungal community whose

abundance and diversity is unaffected by salinity is capable of decomposing particulate OM at the same rate in



salt-affected and non-salt-affected soils. Thus, the decomposition of particulate OM proceeds also in the salt-affected soils, explaining the comparatively low contents of particulate OM and the observation that the proportion of particulate OM was unrelated to salinity.

**Conclusions**

This study aimed at investigating OM dynamics along a salinity gradient in soils of the south-western Siberian Kulunda steppe. Based on previous research, three hypotheses were tested: (i) soil OC stocks decrease along the salinity gradient, because high salinity decreases plant growth and subsequently lowers soil OC inputs, (ii) the proportion and stability of particulate OM is larger in salt-affected soils as compared to non-salt-affected soils as microbial decomposition and transformation of OM is reduced under high salinity levels, and (iii) sodicity

reduces the proportion and stability of mineral-associated OM as the presence of $Na^+$ and a high pH causes dispersion of OM and mineral components. Based on our results, all three hypotheses were rejected. Against our first hypothesis, soil OC stocks increased along the salinity gradient with the most pronounced differences in the topsoil. Contrary to previous studies, plant growth (as determined by above-ground biomass) was not reduced under high salinity levels, suggesting that the soil OC input was similar between salt-affected and non-salt-

affected soils. In contrast to our second hypothesis, the abundance and stability of particulate OM was not related to salinity levels. Remarkably, most of soil OC (>90%) existed in mineral-organic associations (HF >1.6 g $cm^3$) and only a small proportion (<10%) was present in particulate OM (LF <1.6 g $cm^3$). The composition of C isotopes ($\delta^{13}C$, $^{14}C$ activity) and neutral sugars in the density separates suggested a similar degree of OM alteration in salt-affected and non-salt-affected soils. This let´s assume that the microbial activity was not

reduced under high salinity levels. We ascribe this to a functionally diverse and resilient microbial community, as indicated by a fungi : bacteria ratio unaffected by salinity and an even larger fungal PLFA variability under saline conditions, which is capable of decomposing particulate OM at a similar rate in salt-affected and non-salt-affected soils. Contrary to our third hypothesis, the proportion and stability of mineral-bound OM was not reduced under high sodicity levels. High ionic strength of the soil solution fosters the flocculation of soil

constituents and, hence, increases the stability of mineral-organic associations. This, in conclusion, can be the reason for the larger OC stocks in the salt-affected soils: at similar soil OC inputs along the transect and a similar rate of particulate OM decomposition, mineral-associated OM accumulated in the salt-affected soils due to the high ionic strength of the soil solution. In summary, salt-affected soils contribute significantly to the OC storage in the semi-arid soils of the Kulunda steppe. Most of the OC was present in stable mineral-organic associations

and, thus, effectively sequestered in the long-term.

**Acknowledgements**

This study was funded by the Federal Ministry of Education and Research (Germany) in the framework of the Kulunda project (01LL0905). We thank the entire Kulunda team for good collaboration and great team spirit. Acknowledged are Silke Bokeloh, Elke Eichmann-Prusch, Ulrieke Pieper, Fabian Kalks and Michael Klatt for

their reliable assistance in the laboratory. Special thank is dedicated to Leopold Sauheitl for his excellent guidance in the lab.



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



**Tables**

**Table 1: Vegetation (dominant species) and above-ground biomass on each soil type. Given are arithmetic means and the standard error of the mean in parentheses. Significant differences (p <0.05) were not present and are denoted as same lowercase letters.**

| Soil type | Vegetation / dominant species (from most to least dominant) | Above-ground biomass $g\ m^{-2}$ | |
|---|---|---|---|
| Kastanozem | *Festuca valesiaca – Thymus maschallianus – Koeleria glauca* | 164.8 (37.7) | a |
| Non-sodic Solonchak | *Leymus poboanus – Artemisia nitrosa – Atriplex verrucifera* | 133.7 (17.6) | a |
| Sodic Solonchak | *Atriplex verrucifera – Leymus poboanus – Hordeum brevisubulatum* | 139.5 (21.7) | a |





**Table 2: Basic soil parameters as function of soil type and horizon. Given are arithmetic means and the standard error of the mean in parentheses. Abbreviations: n = sample size, BD = bulk density, EC = electrical conductivity, SAR = sodium adsorption ratio, Aggstab = aggregate stability, MWD = mean weight diameter, $Fe_O$ = oxalate-extractable Fe, $Fe_D$ = dithionite-extractable Fe.**

| Soil type | Horizon | n | Depth cm | BD g cm$^{-3}$ | Moisture % of dry weight | pH$_{H2O, 1:2.5}$ - | EC$_{1:5}$ µS cm$^{-1}$ | SAR$_{1:5}$ - | CaCO$_3$ mg g$^{-1}$ | Aggstab ΔMWD (mm) | Clay mg g$^{-1}$ | Silt mg g$^{-1}$ | Sand mg g$^{-1}$ | Fe$_O$ mg g$^{-1}$ | Fe$_D$ mg g$^{-1}$ | Fe$_O$:Fe$_D$ - |
|---|---|---|---|---|---|---|---|---|---|---|---|---|---|---|---|---|
| Kastanozem | Ah | 3 | 23.3 (1.5) | 1.47 (0.07) | 3.6 (0.3) | 7.1 (0.1) | 27 (3) | 0.4 (0.1) | 0 (0) | 0.41 (0.06) | 127 (7) | 230 (20) | 643 (24) | 0.21 (0.20) | 4.9 (0.1) | 0.04 (0.04) |
| | AC | 3 | 48.3 (2.8) | 1.52 (0.07) | 4.5 (0.2) | 8.0 (0.2) | 26 (1) | 0.4 (0.0) | 0 (0) | | 170 (8) | 219 (33) | 611 (35) | 0.16 (0.16) | 4.9 (0.2) | 0.03 (0.03) |
| | Ck | 3 | 114.7 (8.0) | 1.60 (0.07) | 3.6 (0.3) | 8.8 (0.1) | 152 (35) | 0.9 (0.5) | 51 (12) | | 95 (13) | 121 (22) | 784 (35) | 0.04 (0.04) | 3.0 (0.2) | 0.01 (0.01) |
| | C | 2 | 175.0 (15.0) | 1.70 (0.05) | 4.3 (0.4) | 9.0 (0.1) | 236 (101) | 1.7 (0.3) | 29 (1) | | 91 (5) | 125 (22) | 784 (27) | 0.07 (0.07) | 2.9 (0.4) | 0.03 (0.03) |
| Non-sodic Solonchak | Az | 4 | 27.3 (7.1) | 1.44 (0.06) | 20.5 (1.9) | 8.5 (0.2) | 3416 (1053) | 9.6 (2.2) | 53 (16) | 1.02 (0.29) | 174 (14) | 330 (17) | 497 (26) | 0.31 (0.04) | 2.8 (0.7) | 0.13 (0.02) |
| | B | 4 | 62.0 (6.4) | 1.58 (0.02) | 17.8 (1.4) | 8.8 (0.1) | 1378 (372) | 7.0 (0.3) | 102 (28) | | 207 (12) | 313 (21) | 481 (32) | 0.14 (0.07) | 3.7 (0.5) | 0.03 (0.01) |
| | C | 4 | 107.3 (6.1) | 1.78 (0.03) | 14.9 (1.7) | 8.8 (0.1) | 1016 (343) | 5.3 (0.9) | 152 (34) | | 203 (32) | 320 (56) | 477 (87) | 0.07 (0.03) | 3.7 (0.3) | 0.02 (0.01) |
| | Cl | 4 | 175.0 (8.7) | 1.76 (0.03) | 16.5 (0.6) | 8.7 (0.1) | 796 (333) | 3.9 (1.0) | 82 (26) | | 157 (34) | 250 (81) | 593 (114) | 0.24 (0.08) | 3.9 (0.4) | 0.06 (0.02) |
| Sodic Solonchak | Az | 3 | 22.0 (1.5) | 1.23 (0.04) | 30.6 (4.1) | 8.7 (0.1) | 5350 (1476) | 36.0 (10.4) | 207 (22) | 0.33 (0.03) | 192 (55) | 308 (81) | 500 (64) | 0.02 (0.01) | 1.0 (0.3) | 0.02 (0.01) |
| | ACz | 3 | 50.0 (6.1) | 1.29 (0.06) | 29.2 (3.0) | 8.8 (0.0) | 3880 (1590) | 23.8 (8.7) | 264 (22) | | 230 (41) | 307 (45) | 464 (47) | 0.01 (0.00) | 0.9 (0.5) | 0.02 (0.01) |
| | C | 2 | 94.5 (10.5) | 1.65 (0.11) | 20.0 (4.4) | 9.0 (0.1) | 911 (639) | 11.7 (9.7) | 213 (17) | | 190 (34) | 308 (47) | 502 (81) | 0.03 (0.01) | 2.6 (0.3) | 0.01 (0.00) |
| | Cl | 3 | 140.7 (5.2) | 1.78 (0.01) | 16.4 (0.9) | 8.9 (0.0) | 1093 (702) | 8.0 (4.6) | 115 (49) | | 166 (22) | 250 (43) | 584 (60) | 0.32 (0.14) | 3.3 (0.2) | 0.10 (0.05) |



**Table 3: Parameters of OM fractions as function of soil type and horizon. Given are arithmetic means and the standard error of the mean in parentheses. Where n differs for a certain parameter from those indicated in the third column, it is indicated by a separate n in brackets. Abbreviations: OC = organic carbon, MobC = mobilized organic carbon.**

| Soil type | Horizon | n | mg fraction g⁻¹ soil | mg fraction lost (HF) g⁻¹ soil | mg OC g⁻¹ fraction | C : N | δ¹³C (‰) | mg MobC g⁻¹ fraction | % MobC of total OC | % OC of total OC | mg sugar g⁻¹ fraction | mg sugar g⁻¹ OC |
|---|---|---|---|---|---|---|---|---|---|---|---|---|
| | | | **Light fraction (LF)** | | | | | | | | | |
| Kastanozem | Ah | 3 | 5.3 (0.6) | | 119.6 (3.4) | 14.6 (0.4) | -27.48 (0.06) | 17.6 (1.5) [2] | 1.50 (0.05) [2] | 7.30 (0.59) | 46.5 (5.0) | 409.6 (38.2) [1] |
| | AC | 3 | 1.4 (0.1) | | 151.4 (7.0) | 14.2 (0.3) | -27.12 (0.26) | 33.3 (7.4) [2] | 1.25 (0.06) [2] | 4.54 (0.31) | | |
| | Ck | 2 | 0.9 (0.2) | | 218.6 (24.8) | 13.8 (0.8) | -26.42 (0.48) | 75.9 (10.6) | 3.38 (0.89) | 8.29 (2.42) | | |
| Non-sodic Solonchak | Az | 4 | 3.1 (1.1) | | 196.9 (30.8) | 16.7 (1.8) | -26.99 (0.42) | 34.8 (5.3) [2] | 0.71 (0.13) [2] | 3.62 (0.50) | 46.1 - [1] | 328.4 - [1] |
| | B | 4 | 0.9 (0.1) | | 261.4 (14.2) | 17.2 (1.0) | -27.40 (0.28) | 161.0 (13.0) [2] | 1.64 (0.28) [2] | 5.51 (1.08) | | |
| | C | 3 | 0.3 (0.1) | | 279.2 (37.5) | 16.0 (1.1) | -28.08 (0.29) | 236.4 (84.2) [2] | 3.05 (0.52) [2] | 7.26 (0.57) | | |
| Sodic Solonchak | Az | 3 | 4.5 (0.6) | | 265.1 (31.5) | 13.1 (0.9) | -24.27 (1.33) | 46.7 (3.3) [2] | 0.67 (0.18) [2] | 6.91 (2.77) | 104.6 (27.2) | 379.1 (65.2) |
| | ACz | 3 | 1.1 (0.3) | | 246.5 (26.9) | 13.8 (1.0) | -25.11 (0.44) | 130.3 (37.7) [2] | 0.79 (0.20) [2] | 4.18 (2.09) | | |
| | C | 2 | 0.4 (0.1) | | 246.9 (22.4) | 14.7 (1.5) | -26.61 (0.03) | 258.3 (62.7) | 1.93 (0.12) | 4.96 (0.06) | | |
| | | | **Heavy fraction (HF)** | | | | | | | | | |
| Kastanozem | Ah | 3 | 994.7 (0.6) | 7.7 (3.2) | 7.7 (0.3) | 9.1 (0.2) | -23.79 (0.09) | 1.5 (0.1) [2] | 15.56 (0.51) [2] | 92.70 (0.59) | 1.0 (0.2) | 135.6 (22.1) |
| | AC | 3 | 998.6 (0.1) | 28.7 (1.5) | 4.4 (0.3) | 7.5 (0.1) | -23.02 (0.04) | 2.0 (0.4) [2] | 29.41 (1.36) [2] | 95.46 (0.31) | 0.7 (0.1) | 150.7 (15.7) |
| | Ck | 2 | 999.2 (0.2) | 8.8 (6.4) | 2.1 (1.1) | 6.6 (0.7) | -23.04 - [1] | 1.7 (0.1) | 45.71 (12.02) | 91.71 (2.42) | 0.5 - | 171.0 - [1] |
| Non-sodic Solonchak | Az | 4 | 996.9 (1.1) | 85.8 (19.8) | 18.3 (2.7) | 9.8 (0.1) | -23.26 (0.54) | 0.7 (0.1) [2] | 3.72 (0.63) [2] | 96.38 (0.50) | 3.1 (0.6) | 169.3 (27.5) |
| | B | 4 | 999.1 (0.1) | 64.7 (5.6) | 4.7 (0.8) | 8.2 (0.4) | -22.79 (0.36) [3] | 0.2 (0.1) [2] | 5.84 (1.04) [2] | 94.49 (1.08) | 1.0 (0.4) | 171.8 (33.8) |
| | C | 4 | 999.7 (0.1) | 60.7 (6.3) | 2.0 (0.4) | 7.0 (0.3) | -23.04 - [1] | 0.2 (0.1) [2] | 9.43 (1.60) [2] | 92.74 (0.57) [3] | 0.2 - [1] | 136.4 - [1] |
| Sodic Solonchak | Az | 3 | 995.5 (0.6) | 76.4 (14.0) | 19.3 (5.0) | 8.4 (1.7) | -23.40 (0.30) | 0.5 (0.3) [2] | 3.35 (0.95) [2] | 93.09 (2.77) | 5.7 (0.8) | 322.0 (60.8) |
| | ACz | 3 | 998.9 (0.3) | 53.9 (10.0) | 10.6 (2.7) | 10.1 (0.1) | -23.05 (0.13) | 0.1 (0.1) [2] | 2.89 (0.63) [2] | 95.82 (2.09) | 2.6 (0.6) | 244.8 (3.5) |
| | C | 2 | 999.6 (0.1) | 45.8 (4.1) | 3.1 (0.8) | 9.2 (0.1) | -23.20 (0.22) | 0.1 (0.1) | 5.75 (0.38) | 95.04 (0.06) | 0.3 - | 164.8 - |
| | Cl | 1 | 997.2 - | 66.6 - | 1.6 - | 7.9 - | -22.52 - | 0.2 - | | | | |



**Figures**

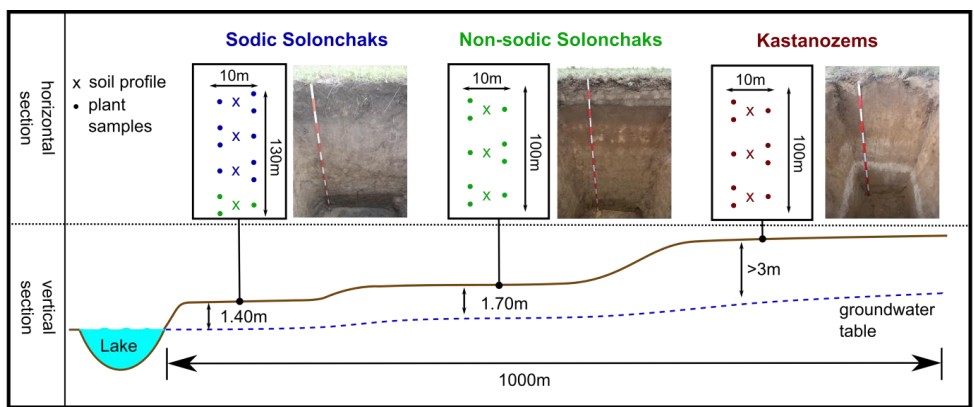

**Figure 1: Schematic representation of study sites and the experimental design. Same colors of the soil profiles and plant samples mark the same soils. A detailed soil type classification of the grouped soils is given in Table S1.**





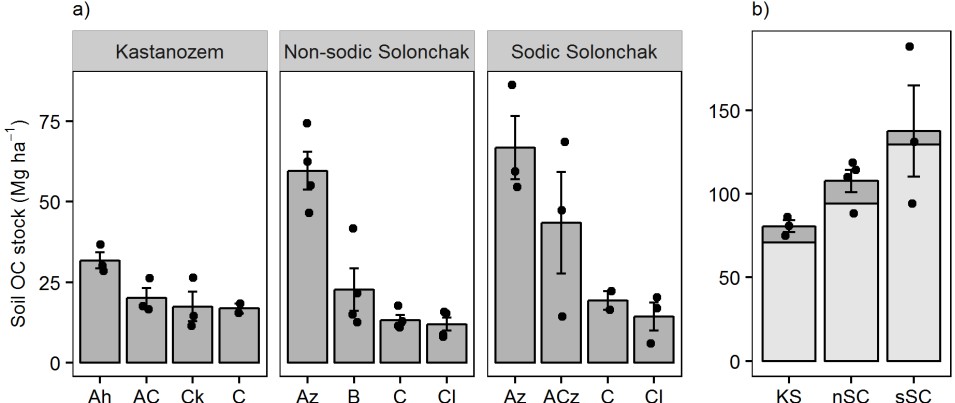

**Figure 2: Soil OC stocks (Mg ha⁻¹) for three soil types, (a) as function of horizon and (b) for a depth of 100 cm and the entire soil profile (light and dark grey). Mean depths of the profiles were 157 ± 20 cm (KS), 175 ± 9 cm (nSC) and 141 ± 5 cm (sSC). Given are arithmetic means ± SE, while dots show individual measurements (in plot b) for the entire soil profile. Abbreviations: KS = Kastanozem, nSC = Non-sodic Solonchak, sSC = Sodic Solonchak.**



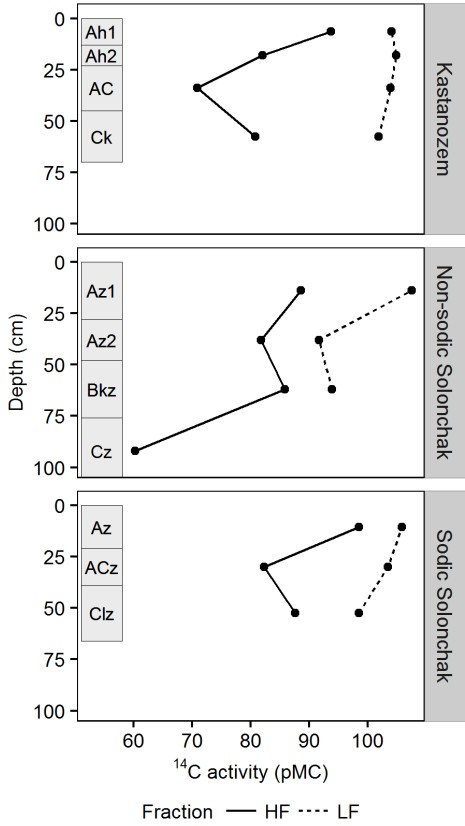

**Figure 3:** $^{14}$**C activity (pMC) for three soil types and two OM fractions as function of soil depth. Rectangles on the left of each panel indicate diagnostic horizons. Abbreviations: LF = light fraction, HF = heavy fraction.**





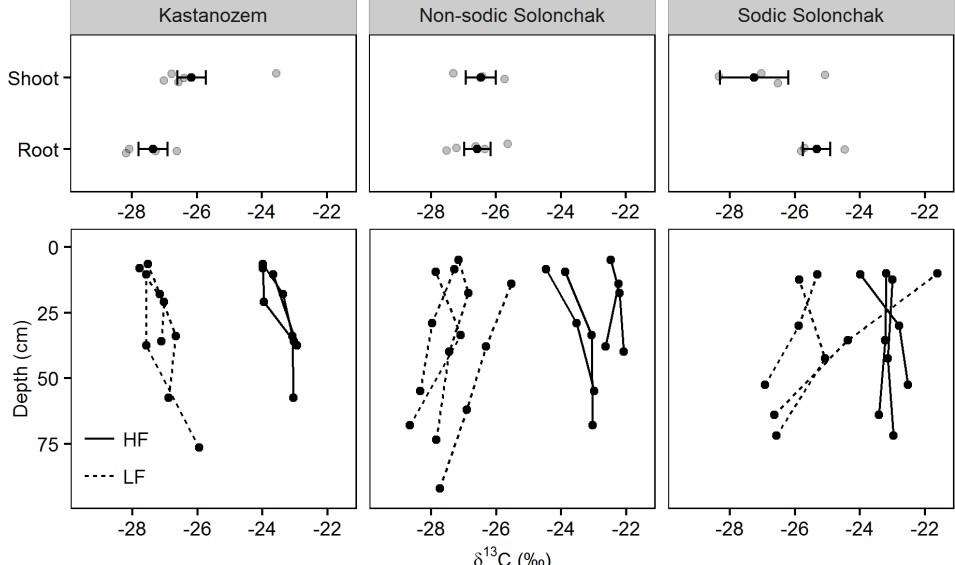

**Figure 4:** δ<sup>13</sup>C ratios of plant components (upper three panels) and of OM present in the light fraction (LF) and the heavy fraction (HF) as function of soil depth (lower three panels) for three soil types. Grey dots in the upper three panels show individual measurements, while the black dots show arithmetic means ± standard error of the mean. In the lower three panels, the three and four replicates per soil type are shown.



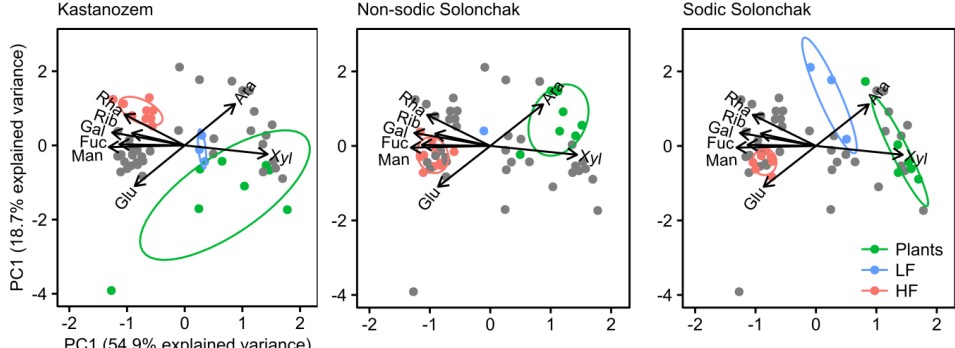

**Figure 5: Biplots derived from a principal components analysis of non-cellulosic neutral sugars from plants, the light fraction (LF) and the heavy fraction (HF), plotted for each soil type separately. The grey dots belong to those samples not considered for the particular soil type. Abbreviations: Man = mannose, Ara = arabinose, Rha = rhamnose, Rib = ribose, Glu = glucose, Fuc = fucose, Xyl = xylose, Gal = galactose.**





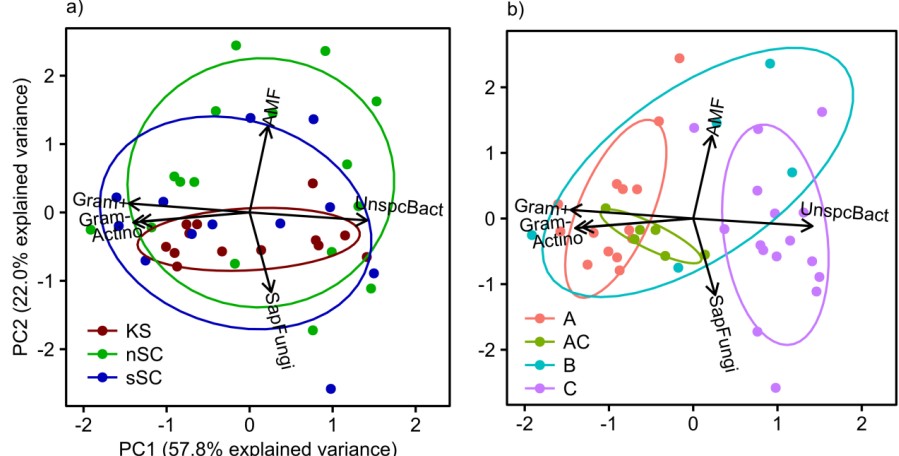

**Figure 6: Biplots derived from a principal components analysis of functional microbial groups as identified from PLFA analysis. Colors and 68% confidence regions are grouped by a) soil type and b) horizon. Abbreviations: KS = Kastanozem, nSC = Non-sodic Solonchak, sSC = Sodic Solonchak, Gram+ = gram-positive bacteria, Gram– = gram-negative bacteria, Actino = actinomycetes, SapFungi = saprotrophic fungi, UnspcBact = unspecific bacteria, AMF = arbuscular mycorrhizal fungi.**