# Peer review of "Organic matter dynamics along a salinity gradient in Siberian steppe soils"

_Biogeosciences, 2017_

## Referee Comment (RC1) · Anonymous Referee #1 · 14 Mar 2017

Bischoff et al. studied soils from a salinity gradient in the Siberian steppe. This salinity gradient covered three different soil types of different salinity and sodicity levels. Along these gradients the authors set out to evaluate three hypotheses: (i) SOC stocks decrease with salinity, (ii) the particulate OM pool is larger and more stable in salt affected soils, (iii) high Na+ concentrations reduce the proportion and stability of mineral-associated OM. To evaluate these hypotheses the authors used samples from different soil horizons and measured SOC and $\delta$13C in the bulk samples as well as in the light and heavy fraction after fractionation. In addition, 14C ages of the fractions were measured, as well as a range of sugars. Microbial community composition was estimated by measuring PLFA patterns. Based on their results, the authors rejected all three of their hypotheses. SOC stocks increased with salinity and the particulate OM fraction was not affected by salinity. Moreover the proportion and stability of mineral-bound OM

was not affected either.

General comments

Considering the increasing extent of salt-affected soils, this MS deals with an important and timely issue. Understanding the carbon dynamics in salt-affected soils is an interesting topic within the scope of Biogeosciences.

The biggest problem of the presented study is that the salinity gradient was confounded by a large difference in soil moisture between the saline sites and the non-saline site. This is discussed by the authors (p. 12 l 37-p.13 l 2), but the importance is understated. Soil moisture has an enormous impact not only on plant productivity, but on decomposition processes, which are inhibited strongly by lack of water (Manzoni et al., 2012). The possibility cannot be excluded that the alteration of OM was found to be similar at saline and non-saline sites, because decomposition was inhibited by lack of water at the non-saline site, especially if the non-saline site is drier than the saline sites throughout the course of the year. It is therefore not possible to conclude that microbial activity was resistant to salinity in the studied soils, since it could have been inhibited by low water availability at all sites, caused by different mechanisms. As a result, a major revision of the discussion is needed. As a suggestion, it could make sense to use water potential as a parameter, to allow for an easier comparison between sites and distinguish between the effects of salinity and moisture.

Another serious issue is that the dataset is very limited, to the extent that statistical hypothesis testing was not possible. Effectively, the number of independent samples along the salinity gradient is only 3.

The manuscript is generally well written, if a bit lengthy in some areas (Results) and underdeveloped in others (discussion). However, there are some sentences that contain clumsy English structures.

Specific comments

p2 l17-19: While you measured the microbial community composition, I do not understand how you derive from the results that the functioning and capability to decompose of the community was virtually unaffected by salt. This seems like an overinterpretation of the data.

p3 l 6-7: This is a bit confusing, since Na+ is also a water-soluble salt. Another issue: Here you refer to Solonchaks and Solonetzes, but later in the MS you switch to sodic and non-sodic Solonchak. Naming should be consistent.

p 3 l 136: which previous studies?

p 4 l 4: What is the expectation for the third objective?

p 4 l. 13-16: As a suggestion, the focus of the MS would become clearer, if the hypotheses would follow your stated objectives above.

p. 8. l. 26: Was plant biomass the only response variable that was tested?

p.8. l.37: By "involved the consideration of several response variables", do you mean multivariate statistics? It is an unclear sentence.

p.9 l. 27-36: This section is never clearly brought up in the discussion and I am not sure if these results contribute important information.

p. 10. l. 21: What could be the reason for decreasing $\delta$13C ratios? Leaching? This is missing a discussion. Could also be linked to the 14C increase with depth.

p. 12 l.9: I don't see any differences in community composition between soil types. Consider changing the wording of "less pronounced".

p. 13 l.16-18: Again, since the Kastanozem was so dry, I would be careful to talk about a lack of inhibition by salinity. Were the OC stocks actually large compared with what would be expected in a steppe soil? Bring this statement into context with data from other studies.

p.13 l. 34-37: How does the salinity in your study compare to that in Peinemann et al. (2005)?

Technical comments:

p. 4. l. 22: Upslope of the lake?

p.9. l. 18: lowest EC1:5. Also in other places in the MS "small" should be replaced by "low", and "large" by "high".

p.10 l.23: Consider changing the order of Figure 3 and 4, so that it matches the first appearences in the text

p. 10 l.36: Did you mean Fig. 3?

p.15 l. 19: This led us to the conclusion….

References

Manzoni, S., Schimel, J.P. and Porporato, A.: Responses of soil microbial communities to water stress: results from a meta-analysis, Ecology, 93 (4), 930-938, 2012.

---

## Short Comment (SC1) · 29 Mar 2017

*A note upfront from the submitting person: This review was prepared by two master students in geography or earth system science at the University of Zurich. The review was part of an exercise during a second semester master level seminar on "the biogeochemistry of plant-soil systems in a changing world", which I organize. We would like to highlight that the depth of scientific knowledge and technical understanding of these reviewers represents that of master students. We enjoyed discussing the manuscript in the seminar, and hope that our comments will be helpful for the authors.*

Rising temperature and anthropogenic influences are the main reason why salt affected soils become more frequent. This study aims to investigate the organic matter dynamics of three different soil types (Kastanozem, non-sodic Solonchak, sodic Solonchak),

along a salinity gradient in the South-Western Siberian Kulunda steppe. Soil samples and the aboveground plants and underground biomass have been characterized by a variety of methods. The results of this study were different from similar studies in the literature, and, and the authors had to reject their initial hypothesis. Surprisingly, organic carbon stocks in the salt-affected were not smaller than in the non-salt-affected soils. Also the abundance and stability of the particulate organic matter was not influenced by salinity. The proportion and stability of mineral-bound organic matter was not reduced under high sodicity levels. Thus, salt-affected soils contribute significantly to the organic carbon storage in the examined region. Also most of the organic carbon was present in stable mineral-organic associations which implies a long-term sequestration.

We liked the readability of the paper. The abstract, the introduction, the discussion and the conclusion are interesting to read. It is a very relevant topic that is important under future climate. However, we had problems to understand the experimental set up. Could the sampling and experimental set up be summarized in a figure or table? Also, for the belowground plant samples we did not understand how they were taken. Were they taken in the profile? Or in about 5 meter distance in every depth, or just once?

As we are only in our second master semester the method section was too long for us. We understand that this section is important for replication. Would it be possible to shorten this section and/or move the details (set up, used instruments, packages, etc.) in the appendix? For non-experts it would help for faster understanding.

We also found many references to figures and tables in the supplement. We are wondering why they are referred to so often, sometimes more often than figures in the the normal text. Could it be, that some figures from the supplement should be Moved back to the main text?

On page 6 in line 3 you the text says "Sample quantity allowed only for two treatments for qualitative analysis" Why are just two treatments for qualitative analysis allowed.

[Figure]

Where there not good enough or to less soil samples? Also on page 11 & 12 in line 20 respectively 13 there was written "data not shown" but for us it was not clear why there are not shown and why you have to state that. If the data are important could you put the data in the supplement?

Table 1: The last column shows "a" but we do not understand why. For table 2 & 3 a line between each soil type would help to read the table. It would also be nice to clarify in the tables itself what the values in parenthesis mean (standard error). The figure 1 was for us quite unclear. We could not make sense of the position in the plant sample dots. Does the position represent on which side they were taken?

Why there are green dots in the Sodic Solonchaks could be stated in the text. However, for us it was not clear. As we wrote above, the experimental set up was mixed with the rest of the text. Not all profiles have the same depth, but this different depth is not represented in the figure. Also in the figure 3 it was for us not that clear why the depth is not the same as in the profiles. In figure 5 a little mistake has slipped in. The y-axis should be PC2 instead of PC1. There we also wondered why the grey dots are not considered as they are quite a lot.

In the conclusion we would also appreciate an outlook for future studies. What would be important to look at?

Some minor comments: - Strange starting sentence of the introduction "... soils... important...." → why do they get more important. They will get more frequent and just to study them will get more important. Maybe "twice as" could be a nicer starting sentence, clearer and nicer as input - Page 3/ line 42 → it is a german sentence; "To date, these soils cover already an area..." do you need "already"? - Page 6/ line 26 → units are at two lines - Page 6/ line 33 → it is written Sect. 2.5, but chapters are not numbered - Page 9/ line 30 ...very broad, peak broadening is related... →you might make two sentences? - Page 15/ line 19 This let's... → informal english

---

## Referee Comment (RC2) · Anonymous Referee #2 · 5 May 2017

This study aimed to understand the role of salinity in shaping soil organic matter. The study is somewhat confounded because the salinity gradient covaries with a moisture gradient. The saltiest soil is closest to the water table and had the highest moisture content while the low salinity soil was far from the water table and had much lower soil moisture. Consequently, it is not possible to separate out the effects of moisture and salinity on the soil carbon and microbial community. Despite this limitation the manuscript presents a robust dataset that is, on the whole, well contextualized.

The presentation of the data is quite dense and the manuscript is made less comprehensible by the excessive use of abbreviations. The authors should work to simplify the results where data is sometimes redundantly presented in the text, tables and figures. There also seems to be an excess of supplemental data that is simply an alternate presentation of the data shown in the tables.

[Figure]

Generally, I think the authors could do more to explain why their findings do not match those reported by others. The moisture gradient seems to be the most obvious reason to me yet this is not well discussed in the manuscript.

Page 3 - 31. Suggest start new sentence, i.e. change "OM, while particulate OM" to "OM. In contrast, particulate OM"

Page 8 - 5 If the salt interfered with the internal standard peak how can you be sure it did not interfere with any of the other peaks?

Figure 3 is not referred to in the results section

Note to self salty soils have more clay and more moisture – these are factors that stabilize C

Page 10 - 25 Can you write out SPT this is not used frequently enough to warrant abbreviation

Page 10 -27 Can you just refer to the loss as mobilized C, I think that would make it less confusing. I had to reread the methods to understand this part of the results.

Page 10- 32 I think this is a sentence for the discussion.

Page 10 – 37 What does B.P. stand for ? Before Present?

Figure 4 is also not referred to in the results- only the tables. Perhaps the data should not be redundantly presented in both locations?

Figure 5 – Is there a need to show the grey dots in each panel?

Page 11 – 21-33 Have you considered doing a PerMANOVA to determine if these differences in sugar composition are significant?

Page 11 – 35 this sentence is confusing " The relative contribution of PLFA observed within the PLFA profiles "

Page 12 – 5 As with sugar composition you should be able to test statistically if the

sites and soil profiles are statistically distinct in terms of microbial community structure.

Pag 12- 20 Given the high CV for these soil types I'm not sure that soil type is such a great predictor of carbon stock.

Page 13 – 14 How are you sure the soils are not affected by erosion?

Page 13-15 – Could reduced decomposition due to salt stress and anaerobic conditions from the high moisture content be contributing to the higher organic matter content in the Non-sodic and sodic Solonchaks?

Page 13 – 30 Could you remind us what your second hypothesis was?

Page 13 – it seems that the water availability to plants and microbes might be similar in the dry salt free Kastanozem and the wetter but salty Solonchaks (i.e. similar osmotic pressure). This could explain why above ground biomass was similar and explain the similarities in soil C.

Page 15-19 this lets us assume?

––––––––––––––––––––––––––––––

---

## Author Comment (AC1) · 7 Jun 2017

Dear Referee #1 (R#1),

Thank you for taking your time to go through our manuscript and give critical comments and advices. We agree with you that we partially overinterpreted the results (deriving from PLFA measurements the functionality and resilience of the microbial community) and that we should discuss some data more carefully. After discussing your comments we decided to add a fourth hypothesis to the existing three: (i) soil OC stocks decrease with increasing salinity, (ii) the proportion and stability of particulate OM is larger in salt-affected than in non-salt-affected soils, (iii) sodicity reduces the proportion and stability of mineral-associated OM, and (iv) fungi : bacteria ratios, as derived from

PLFA measurements, decrease along the salinity gradient. By that, we connect our objectives with the hypotheses, as you suggested. Moreover, we are not going to relate the PLFA data directly to the proportion and stability of particulate OM, as this in fact would result in an overinterpretation. Setting up a fourth hypothesis allows for a separate discussion of the microbial community data. We are going to discuss the possibility that a functionally diverse fungal community contributed to the progressive decomposition of particulate OM. In addition, we will clearly state that a similar water stress along the salinity gradient could be responsible that we have not found a different alteration of OM along the transect. In the introduction we will remove the explanation on Solonetz soils, but focus on salinity and sodicity, as these were the main issues of our study. By that we follow many of your helpful advices. We are going to await the decision of the editor and, if positive, work on the manuscript revision.

General comments

R#1: Considering the increasing extent of salt-affected soils, this MS deals with an important and timely issue. Understanding the carbon dynamics in salt-affected soils is aninteresting topic within the scope of Biogeosciences. The biggest problem of the presented study is that the salinity gradient was confoundedby a large difference in soil moisture between the saline sites and the non-saline site.This is discussed by the authors (p. 12 l 37-p.13 l 2), but the importance is under stated. Soil moisture has an enormous impact not only on plant productivity, but ondecomposition processes, which are inhibited strongly by lack of water (Manzoni et al.,2012). The possibility cannot be excluded that the alteration of OM was found to besimilar at saline and non-saline sites, because decomposition was inhibited by lack ofwater at the non-saline site, especially if the non-saline site is drier than the saline sitesthroughout the course of the year. It is therefore not possible to conclude that microbialactivity was resistant to salinity in the studied soils, since it could have been inhibitedby low water availability at all sites, caused by different mechanisms. As a result, amajor revision of the discussion is needed. As a suggestion, it could make sense touse water potential as a parameter,

to allow for an easier comparison between sitesand distinguish between the effects of salinity and moisture.

Authors (A):We agree that the salinity gradient was possibly confounded by a difference in soil moisture. Hence, the interpretation of the data is currently not straightforward. After evaluating the data for the first time, we were aware of the problem and intended to calculate the water potential of the soils, as you have suggested. Thereby, we faced the problem, that we could not measure the matric potential directly by use of soil water retention curves, since we had no undisturbed soil cores of the studied soils. Thus, we had to use Pedo-Transfer-Functions (PTF's) which estimate the matric potential via soil parameters, such as soil texture, bulk density, organic carbon content, and actual soil water content. Such a PTF was proposed in Vereecken et al. (1989) with the Van Genuchten model. Another possibility is to calculate the model parameters for the Van Genuchten model via the software "RETC". Both, the use of the PTF's in Vereecken et al. (1989) and the use of "RETC" have not yielded plausible results for our soils. This might be due to the fact, that the PTF's were empirically developed for temperate soils without influence of salinity. As a consequence, we cannot calculate the matric potential for the soils in our study and, thus, neither the water potential. However, this is not limiting the significance of our study, since (i) it is a natural phenomenon that salinity co-varies with soil moisture in the study area, thus, our transect represents the occurring natural conditions, (ii) the soil moisture measurement given in Table 2 represents just a "snapshot" at the moment of soil sampling and not a mean value during a longer period of time. To draw conclusions about the possible effect of the matric potential or water potential, respectively, on processes like soil OM decomposition, we would need to measure these parameters for a longer period of time. Nevertheless, we agree with you that the effect of soil moisture is a critical aspect in the manuscript and should be discussed more extensively. In the revised manuscript we intend to revise the discussion thoroughly, particularly with respect to the effect of matric potential vs. osmotic potential and the overall water potential on soil OM decomposition. In particular we are going to state that it is possible that we have not found differences between the

soils with respect to soil OM decomposition, because of a similar water stress/water potential in all soils.

R#1: Another serious issue is that the dataset is very limited, to the extent that statistical hypothesis testing was not possible. Effectively, the number of independent samplesalong the salinity gradient is only 3.

A: The number of independent samples along the salinity gradient was 3 or 4 (for the Non-sodic Solonchaks), respectively. Thus, statistical hypothesis testing was not possible, as noted on p. 8 l. 34Âň-36 of the manuscript. However, this does not mean that the dataset is limited. We decided to conduct an in-depth analysis by measuring many soil parameters per soil profile and relate them to each other in order to reveal processes which take place within the soil. This was done in many previous studies (Fierer et al., 2003; Kemmitt et al., 2008; Shen and Bartha, 1996). By that, we actually obtained a very large and detailed data set. For example, only by use of isotopic data (13C, 14C) and neutral sugar measurements in combination with PLFA we could reveal that POM was not distinctly altered in the studied soils, maybe due to a functionally diverse and resilient microbial community, which is capable of decomposing POM at a similar rate in salt-affected and non-salt-affected soils. As you have mentioned in the previous comment, in the revised manuscriptwe will add to this explanation, that it is possible that a lower soil moisture in the non-salt-affected soils has led to similar POM decomposition in the salt-affected and non-salt-affected soils.

R#1: The manuscript is generally well written, if a bit lengthy in some areas (Results) and underdeveloped in others (discussion). However, there are some sentences that containclumsy English structures.

A: In the revised manuscript we are going to shorten the results section (e.g. the part about soil mineralogy and by generally not repeating the numbers from the tables too extensively). On the other hand, we are going to work on the discussion section including more detail and discussing also controversial positions, such as the fact that

soil moisture could have a crucial impact on soil OM decomposition along the transect. Sentences that contain clumsy English structures are going to be revised.

Specific comments

R#1: p2 l17-19: While you measured the microbial community composition, I do not understand how you derive from the results that the functioning and capability to decomposeof the community was virtually unaffected by salt. This seems like an overinterpretationof the data.

A: As mentioned above, we agree that this could be an overinterpretation of the data. We are going to soften this conclusion in the revised manuscript.

R#1: p3 l 6-7: This is a bit confusing, since Na+ is also a water-soluble salt. Another issue:Here you refer to Solonchaks and Solonetzes, but later in the MS you switch to sodicand non-sodic Solonchak. Naming should be consistent.

A: Na+, as such, is not a water-soluble salt but a monovalent cation. To make the sentence clearer, we may change it in the revised manuscript to: "Solonchaks contain high loads of water-soluble salts in general, while Solonetzes are particularly characterized by Na+ as the dominant cation on the exchange sites, irrespective of the quantity of salts." Here we distinguish between Solonchak and Solonetz to explain the difference between non-sodic and sodic. But, we agree with you, that we could shorten the explanation regarding "Solonetz" in the revised manuscript, as this particular soil type was not part of our transect.

R#1: p 3 l 136: which previous studies?

A:Thank you for this attentive note. Previous studies are for example Mavi et al. (2012), Setia et al. (2013, 2014). We are going to add this to the revised manuscript.

R#1: p 4 l 4: What is the expectation for the third objective?

A: After your comment about the "overinterpretation" of the PLFA data (microbial community composition / functioning), we decided to attenuate the conclusion on the results of the third objective. Moreover, we came to the conclusion that it is not straightforward to relate the PLFA data to the results of soil OC stocks and quantities and properties of functionally different OM fractions. Thus, we will restate our third objective to "(iii) analyse changes of the microbial community composition". This objective will be kept quite general, as to our knowledge there are no studies which have determined microbial community compositions in Solonchaks or Solonetzes so far (which we stated on p. 3 l. 34-36). We are going to include this in the revised manuscript.

R#1: p 4 l. 13-16: As a suggestion, the focus of the MS would become clearer, if the hypotheses would follow your stated objectives above.

A: In the revised manuscript we are going to integrate your suggestion. We will set up three objectives and add a fourth hypothesis regarding the microbial community composition.

R#1: p. 8. l. 26: Was plant biomass the only response variable that was tested?

A: Yes, plant biomass was the only response variable that was tested, because this was the only parameter for which we had a sufficient number of samples/replicates. This was because it is a parameter which is easy to measure without the need of lots of time and money.

R#1: p.8. l.37: By "involved the consideration of several response variables", do you mean multivariate statistics? It is an unclear sentence.

A:In the revised manuscript we will change the sentence to: "Data of PLFA and neutral sugars were analyzed by PCA in order to consider multiple response variables. Confidence regions (68%) for the group centroids of the independent factor variables were added to the biplots."

R#1: p.9 l. 27-36: This section is never clearly brought up in the discussion and I am not sure if these results contribute important information.

A:We determined the soil mineralogical composition principally because of two reasons: (i) to characterize the mineralogical composition of water-soluble salt minerals in the salt-affected soils, and (ii) to determine the clay mineralogical composition particularly with respect to expandable clay minerals, such as smectite, as these affect the physical properties of sodic soils crucially (see p. 6 l.4-5). The mineralogical characterization of the water-soluble salt minerals is primarily descriptive, but informative and important as we study salt-affected soils. The clay mineralogical composition turned out to be similar between the soils and therefore cannot explain differences between the soils later on in the discussion. Thus, we may move this section to the Supplements in the revised manuscript.

R#1: p. 10. l. 21: What could be the reason for decreasing _13C ratios? Leaching? This is missing a discussion. Could also be linked to the 14C increase with depth.

A: In our opinion, decreasing d13C ratios cannot be caused by leaching as the net-movement of water in the salt-affected soils is upwards. Decreasing d13C ratios, and as such increasing 14C activities, with depth could be related to a faster soil OM turnover. In the Solonchaks of our study this could occur due to the water stress in the topsoil (osmotic stress and matric stress) while the subsoil is generally wetter due to the proximity to the groundwater and a lower salt content. Hence, the conditions for microbes to process soil OM could be better in the subsoil than in the topsoil. This would explain the observed pattern in the Solonchak, but not the increase of 14C activity in the Kastanozem. Since this is very speculative, we decided to leave it out from the discussion. But we may add it to the revised manuscript with the advice that this asks for further investigation in future studies.

R#1: p. 12 l.9: I don't see any differences in community composition between soil types. Consider changing the wording of "less pronounced".

A: Indeed, there are differences in the microbial community composition between the soil types. Please consider the confidence regions in Fig. 6a with a larger variability on

PC2 for the salt-affected soils. This corresponds to a larger variability of fungal PLFA in the salt-affected soils. Though the differences are small, they are existent and should be mentioned. However, we are going to change the wording to "small" instead of "less pronounced" in the revised manuscript.

R#1: p. 13 l.16-18: Again, since the Kastanozem was so dry, I would be careful to talk about a lack of inhibition by salinity. Were the OC stocks actually large compared with whatwould be expected in a steppe soil? Bring this statement into context with data fromother studies.

A: As mentioned in a previous response to one of your comments, in the revised manuscript we are going to include a more intensive discussion on the fact that the very dry conditions in the Kastanozem could have led to a similar water stress in the Kastanozems and Solonchaks, with the respective consequences on soil OM input and soil OM decomposition. So far this was only little discussed in the manuscript (p. 12 l. 37-39, p. 13 l. 1-5). As already mentioned in the manuscript, the OC stocks of Solonchaks were large when compared to data from other studies, while the Kastanozems of the transect revealed smaller OC stocks than previously observed in other studies (see chapter "Discussion: Soil OC stocks along the salinity gradient").

Technical comments:

R#1: p. 4. l. 22: Upslope of the lake?

A: "to about 5m above the lake"

R#1: p.9. l. 18: lowest EC1:5. Also in other places in the MS "small" should be replaced by"low", and "large" by "high".

A: Thank you for this correction. We are going to correct this in the revised manuscript.

R#1: p.10 l.23: Consider changing the order of Figure 3 and 4, so that it matches the first appearances in the text

A:In the revised MS, we are going to change the order of Figure 3 and 4.

R#1: p. 10 l.36: Did you mean Fig. 3?

A: Correct. We are going to change this in the revised MS.

R#1: p.15 l. 19: This led us to the conclusion.

A: Thanks for the correction. We are going to integrate this in the revised MS.

References:

Fierer, N., Schimel, J. P. and Holden, P. A.: Variations in microbial community composition through two soil depth profiles, Soil Biol. Biochem., 35(1), 167–176, doi:10.1016/S0038-0717(02)00251-1, 2003.

Kemmitt, S. J., Lanyon, C. V., Waite, I. S., Wen, Q., Addiscott, T. M., Bird, N. R. a, O'Donnell, a. G. and

Brookes, P. C.: Mineralization of native soil organic matter is not regulated by the size, activity or composition of the soil microbial biomass-a new perspective, Soil Biol. Biochem., 40, 61–73, doi:10.1016/j.soilbio.2007.06.021, 2008.

Mavi, M. S., Sanderman, J., Chittleborough, D. J., Cox, J. W. and Marschner, P.: Sorption of dissolved organic matter in salt-affected soils: effect of salinity, sodicity and texture., Sci. Total Environ., 435–436, 337–44, doi:10.1016/j.scitotenv.2012.07.009, 2012.

Setia, R., Rengasamy, P. and Marschner, P.: Effect of exchangeable cation concentration on sorption and desorption of dissolved organic carbon in saline soils, Sci. Total Environ., 465, 226–232, doi:10.1016/j.scitotenv.2013.01.010, 2013.

Setia, R., Rengasamy, P. and Marschner, P.: Effect of mono- and divalent cations on sorption of water-extractable organic carbon and microbial activity, Biol. Fertil. Soils, 50(5), 727–734, doi:10.1007/s00374-013-0888-1, 2014.

Shen, J. and Bartha, R.: Metabolic efficiency and turnover of soil microbial communities in biodegradation tests., Appl. Environ. Microbiol., 62(7), 2411–2415, 1996.

Vereecken, H., Maes, J., Feyen, J. and Darius, P.: Esitmating the soil moisture retention chracteristic from texture, bulk density, and carbon content, Soil Sci., 148(6), 389–403, doi:10.1097/00010694-198912000-00001, 1989.

---

## Author Comment (AC2) · 7 Jun 2017

Dear Referee #2 (R#2),

Thank you for taking your time to go through our manuscript and give us helpful advices and corrections.

R#2: This study aimed to understand the role of salinity in shaping soil organic matter. The study is somewhat confounded because the salinity gradient covaries with a moisture gradient. The saltiest soil is closest to the water table and had the highest moisturecontent while the low salinity soil was far from the water table and had much lowersoil moisture. Consequently, it is not possible to separate out the effects of moistureand salinity on the soil carbon and microbial community. Despite this limitation

themanuscript presents a robust dataset that is, on the whole, well contextualized.The presentation of the data is quite dense and the manuscript is made less comprehensible by the excessive use of abbreviations. The authors should work to simplify theresults where data is sometimes redundantly presented in the text, tables and figures. There also seems to be an excess of supplemental data that is simply an alternatepresentation of the data shown in the tables. Generally, I think the authors could do more to explain why their findings do not match those reported by others. The moisture gradient seems to be the most obvious reason to me yet this is not well discussed in the manuscript.

A: It is true that the salinity gradient covaries with a moisture gradient, which is a broadly occurring situation in the study area. Salt-affected soils are those close to the groundwater table and are thus generally moister than the non-salt-affected soils which occur at a larger distance to the groundwater table. Therefore, this should not be seen as a limitation of the study. It is just the natural association of the soils in the semi-arid steppe. We agree with you that we should discuss this in more detail in the revised manuscript. In particular we are going to include the fact that it is difficult to separate the effect of salinity and moisture on soil OM dynamics and the microbial community. We will discuss that the missing effect of salinity on the soil OM dynamics along the transect could also be explained by the covarying moisture gradient. According to your comment we are going to delete some abbreviations, e.g. SPT. If data is redundantly presented, we will delete redundant data, e.g. the d13C ratios in Table 3 which are similarly presented in Figure 4. In our opinion there is no excess of supplemental data and it is not an alternate presentation of the data shown in the tables. Only Figure S5 and S6 are partially redundant to Figure 5, but we consider these figures as informative as they highlight precisely how the two sugars glucose and arabinose differ between the soil types with respect to plant samples and organic matter fractions. This is not possible to show in a PCA in such detail. In the revised manuscript we are going to explain in more detail why we think that our results differ from those observed in other studies. As was already discussed, this could be due to the covarying moisture gradient, particularly the pronounced aridity in the Kastanozem, which may have led to a smaller OM input and consequently smaller OC stocks. Moreover, soil OM decomposition could have been inhibited and thus the soil OM transformation appears similar to those of the salt-affected soils.

R#2:Page 3 - 31. Suggest start new sentence, i.e. change “OM, while particulate OM” to“OM. In contrast, particulate OM”

A:Thank you for this advice. We are going to change it accordingly in the revised manuscript.

R#2: Page 8 - 5 If the salt interfered with the internal standard peak how can you be sure it did not interfere with any of the other peaks?

A: We are very sure that the salt did not interfere with any of the other PLFA peaks, as the peaks appeared clear and with a characteristic shape. Moreover, the shape and the appearance of the peaks were similar in the salt-affected and non-salt-affected soils. This was not the case for the internal standard peak, which clearly differed in the topsoils of the salt-affected soils by a clear overlap with another unspecific peak.

R#2: Figure 3 is not referred to in the results section

A: By accident, on p.10 l. 36 we referred to “Table 3” instead of “Figure 3”. We are going to change this in the revised manuscript.

R#2: Note to self salty soils have more clay and more moisture – these are factors thatstabilize C

A: To argue that a higher clay content may contribute to a larger C stabilization in the salt-affected soils is a good point. We are going to integrate this in the revised manuscript. To discuss soil moisture as a factor that stabilizes C along the transect is, in our opinion, difficult, because the “available” soil moisture (i.e. water potential) is possibly similar along the transect, either due to a low osmotic potential (Solonchaks) or a low matric potential (Kastanozems). Thus, we would not like to integrate this into

the discussion.

R#2: Page 10 - 25 Can you write out SPT this is not used frequently enough to warrant abbreviation

A: Yes, we are going to write it out in the revised manuscript.

R#2: Page 10 -27 Can you just refer to the loss as mobilized C, I think that would make itless confusing. I had to reread the methods to understand this part of the results.

A:We think it would not be correct to refer to the loss simply as mobilized C, since the mass loss in the salt-affected soils is largely due to the dissolution of C-free salts. Hence, in these soils we observe a large mass loss but only a minor loss of mobilizable C. In the Kastanozem, however, less total soil mass was lost during the density fractionation but this was associated with a larger portion of mobilized OC. Since we think that this differentiation is important, we would like to keep it in our revised manuscript.

R#2: Page 10- 32 I think this is a sentence for the discussion.

A: This is what we explained in the previous comment. We agree with you, that this sentence fits better into the discussion. Hence, we are going to move it in the revised manuscript.

R#2:Page 10 – 37 What does B.P. stand for ?Before Present?

A: Yes, B.P. means "Before Present".

R#2: Figure 4 is also not referred to in the results- only the tables. Perhaps the data should not be redundantly presented in both locations?

A: Indeed, figure 4 is referred to in the results (p. 10 l. 23). However, we agree that the data on d13C is somewhat redundant. In the revised manuscript we are going to delete d13C values from Table 3 and refer only to Figure 4.

R#2: Figure 5 – Is there a need to show the grey dots in each panel?

A: The PCA on neutral sugars was applied on the entire data set, i.e. neutral sugar data of all three soil types and all three fractions was analyzed in one PCA. This resulted in the biplot shown in Figure 5. To highlight differences between the soils we split the biplot into three panels and indicated the fractions of each soil by different colors. The biplot shows all considered data (i.e. the entire data set); this includes the grey dots which do not belong to the particular soil type of a panel.

R#2: Page 11 – 21-33 Have you considered doing a PerMANOVA to determine if these differences in sugar composition are significant?

A: PerMANOVA is a robust tool to test multivariate data on statistical significance. However, a minimum sample size is required to obtain reliable results. As mentioned in the statistics section of the Material & Methods part, we only have 3Âň–4 field replicates (i.e. 3–4 soil profiles per soil type). One might argue that we consider more than 3–4 samples per soil type in the PCA, but this is because the soil profiles were sampled in horizons and data of each horizon is also considered in the PCA. As these horizon samples are nested within the soil profile they cannot be treated as independent samples. If so, they would be referred to as "pseudo-replicates" and application of statistical hypothesis testing on such data would result in underestimation of p-values. Therefore we decided to refrain from a PerMANOVA and analyze the data descriptively.

R#2: Page 11 – 35 this sentence is confusing " The relative contribution of PLFA observed within the PLFA profiles"

A: In the revised manuscript we are going to change the sentence to: "The relative proportion of PLFA on the entire data set was as follows:"

R#2:Page 12 – 5 As with sugar composition you should be able to test statistically if thesites and soil profiles are statistically distinct in terms of microbial community structure.

A: Please refer to our previous comment regarding statistical hypothesis testing on

neutral sugar data.

R#2: Pag 12- 20 Given the high CV for these soil types I'm not sure that soil type is such a great predictor of carbon stock.

A: Soil type is, for sure, not the best predictor of C stocks but it is, as a single variable, possibly more precise than solely predictors such as temperature, moisture, parent material, or clay content. Anyway, this was not the reason why we discuss C stocks of soil types in that part of the manuscript. In this section we aim to compare our measured data on C stocks to data of previous studies investigating similar soils. This is particularly important as our data is different from what was found in other studies. We therefore would like to keep this part in the revised manuscript.

R#2: Page 13 – 14 How are you sure the soils are not affected by erosion?

A:This is explained on p.4 l.30. Sample locations were plane with <0.5° slope inclination. Thus, the probability of erosion is reduced to a minimum.

R#2: Page 13-15 – Could reduced decomposition due to salt stress and anaerobic conditionsfrom the high moisture content be contributing to the higher organic matter content inthe Non-sodic and sodic Solonchaks?

A: We would expect that a reduced decomposition due to salt stress would result in an accumulation of particulate OM. This was not the case in the studied soils: salt-affected and non-salt-affected soils contained similar proportions of particulate OM. Moreover, the analysis of C isotopes and neutral sugars indicated a comparable degree of OM alteration between the soils, as already discussed in the manuscript, while we would expect a smaller OM alteration if decomposition would be reduced in the salt-affected soils. Higher OC stocks in the salt-affected soils were particularly found in the topsoils. Anaerobic conditions in the topsoil are very unlikely as the gleyic properties of the soils show that anaerobic conditions can be primarily expected in the subsoil, but not in the topsoil. For example, in the four soil profiles next to the lake, Fe and Mn mottling

reaches on average 84 ± 16 cm soil depth, thus indicating the maximum average ground water level during flooding. In the subsoil differences between OC stocks were smaller. Thus, we do not consider anaerobic conditions as a factor explaining the high OC stocks in the Solonchaks.

R#2: Page 13 – 30 Could you remind us what your second hypothesis was?

A: In the revised manuscript we are going to repeat the second hypothesis at the beginning of that paragraph.

R#2: Page 13 – it seems that the water availability to plants and microbes might be similar inthe dry salt free Kastanozem and the wetter but salty Solonchaks (i.e. similar osmotic pressure). This could explain why above ground biomass was similar and explain the similarities in soil C.

A: This is a good point and was already noted in the manuscript (p. 12 l. 37 – p. 13 l. 5). However, we think we should discuss this issue more intensively in the revised manuscript. This was already noted by Referee #1 and we are going to revise the discussion in the revised manuscript with particular focus on that point.

R#2: Page 15-19 this lets us assume?

A:We are going to change that in the revised MS to "This led us to the conclusion. . ."

---

## Author Comment (AC3) · 7 Jun 2017

Dear M.W.I Schmidt, Dear master students,

Thank you for taking your time to discuss our manuscript and give advices for improvements.

Short comment #1 (SC #1): *A note upfront from the submitting person: This review was prepared by two master students in geography or earth system science at the University of Zurich. The review was part of an exercise during a second semester master level seminar on "the biogeochemistry of plant-soil systems in a changing world", which I organize. We would like to highlight that the depth of scientific knowledge and technical understanding of these reviewers represents that of master students. We enjoyed

discussing the manuscript in the seminar, and hope that our comments will be helpful for the authors.*

Rising temperature and anthropogenic influences are the main reason why salt affectedsoils become more frequent. This study aims to investigate the organic matter dynamicsof three different soil types (Kastanozem, non-sodic Solonchak, sodic Solonchak), along a salinity gradient in the South-Western Siberian Kulunda steppe. Soil samples and the aboveground plants and underground biomass have been characterized by avariety of methods. The results of this study were different from similar studies in the literature,and, and the authors had to reject their initial hypothesis. Surprisingly, organiccarbon stocks in the salt-affected were not smaller than in the non-salt-affected soils. Also the abundance and stability of the particulate organic matter was not influenced by salinity. The proportion and stability of mineral-bound organic matter was not reduced under high sodicity levels. Thus, salt-affected soils contribute significantly to the organic carbon storage in the examined region. Also most of the organic carbon was present in stable mineral-organic associations which implies a long-term sequestration. We liked the readability of the paper. The abstract, the introduction, the discussion and the conclusion are interesting to read. It is a very relevant topic that is important under future climate. However, we had problems to understand the experimental setup. Could the sampling and experimental set up be summarized in a figure or table?

Authors (A): Thank you for this evaluation of our manuscript.We are going to explain the experimental setup more clearly in the revised manuscript, particularly the part on p. 4 l. 31-39 will become changed. However, please note that we included already a figure explaining the experimental setup in the existing manuscript (Figure 1).

SC #1: Also, for the belowground plant samples we did not understand how they were taken.

A: To characterize the isotopic composition (d13C) and neutral sugars of plant samples,

we retrieved whole plants of the dominant plant species (see Table 1) from the soil. Subsequently, we split the plant into two parts: roots and shoots.

SC #1: Were they taken in the profile? Or in about 5 meter distance in every depth, or justonce?

A: With respect to plant samples, we took three replicates in about 5m distance to the profile. This is explained on p. 4 l. 38 – p. 5. l.2 and also shown in Figure 1.

SC #1: As we are only in our second master semester the method section was too long forus. We understand that this section is important for replication. Would it be possible toshorten this section and/or move the details (set up, used instruments, packages, etc.) in the appendix? For non-experts it would help for faster understanding.

A: We agree that the method section is very long. But this is owed to the many methods we used to collect our data. Methods like density fractionation, neutral sugars analysis or PLFA have to be explained in such detail. Also other methods, as the determination of OC, TN, and d13C, are non-trivial and deserve a paragraph of explanation. However, we decided to move the part about soil mineralogical composition into the supplement of the revised MS as it does not contribute substantial data which is discussed later on.

SC #1:We also found many references to figures and tables in the supplement. We are wondering why they are referred to so often, sometimes more often than figures in the the normal text. Could it be, that some figures from the supplement should be Moved backto the main text?

A:The supplemental data (figures and tables) give additional information which contribute to the understanding of the manuscript but are not necessary for a deep discussion of the data. Hence, we would like to keep it as is and not move part of it into the main text of the MS.

SC #1:On page 6 in line 3 you the text says "Sample quantity allowed only for two treatments for qualitative analysis" Why are just two treatments for qualitative analysis

allowed. Where there not good enough or to less soil samples?

A: In XRD analysis there are usually four treatments used to distinguish the clay mineralogical composition of a soil sample: (i) $Mg^{2+}$-saturation, (ii) $Mg^{2+}$ + ethylene glycol saturation, (iii) $K^+$-saturation and (iv) $K^+$-saturation + heating to 550K. We had not enough sample mass to conduct all four treatments, thus we had to decide for two of the treatments. As we were interested in the quantity of expandable (swelling) clay minerals such as smectite, we decided to use the "standard" treatment ($Mg^{2+}$-saturation) and the $Mg^{2+}$ + ethylene glycol saturation, as the combination of both yields the necessary results.

SC #1:Also on page 11 & 12 in line 20 respectively 13 there was written "data not shown" but for us it was not clear why thereare not shown and why you have to state that. If the data are important could you putthe data in the supplement?

A: On p. 11 l. 20 we state the relative proportion of each neutral sugar on the entire data set. This is to give an overview to the data and not necessary to repeat in a table. Otherwise it would be a redundant presentation of data. On p. 12 l. 13 we write about fungi : bacteria ratios. We agree with you that it would be informative to the reader if we add the data to the supplements.

SC #1:Table 1: The last column shows "a" but we do not understand why.

A: As is noted in the heading of the table, these letters indicate whether there are significant differences between the samples or not.

SC #1:For table 2 & 3 aline between each soil type would help to read the table. It would also be nice to clarifyin the tables itself what the values in parenthesis mean (standard error).

A: We are going to add a line between soil types for better readability. We already clarified in the heading of the table the meaning of the value in parenthesis (standard error).

SC #1: The figure 1 was for us quite unclear. We could not make sense of the position in the plant sample dots. Does the position represent on which side they were taken? Why there are green dots in the Sodic Solonchaks could be stated in the text. However,for us it was not clear. As we wrote above, the experimental set up was mixed with therest of the text. Not all profiles have the same depth, but this different depth is notrepresented in the figure.

A: Yes, the dots represent the approximate position where the samples were taken. This is also indicated by the arrows which highlight the distance of the sampling locations to each other. As stated on p. 4 l. 32-36, four soil profiles were analyzed on the foot slope of the transect because of the larger site heterogeneity there. However, laboratory analyses afterwards revealed that one of the four soils was not sodic and had to be grouped together with the non-sodic Solonchaks. This exactly is shown in Figure 1. We also explained the meaning of the colors in Figure 1. The different depth of the groundwater table, which resulted in different depths of the soil profiles, is clearly shown in Figure 1.

SC #1: Also in the figure 3 it was for us not that clear why the depthis not the same as in the profiles.

A: 14C analyses are very costly and to measure all samples of a profile was therefore not possible for us. We therefore decided to measure all samples until the topmost C horizon of a profile, because the topmost horizons are those with the highest OC contents. Moreover, in the topmost horizons we observed the largest differences between the soils with respect to their OC stocks. Only in the Non-sodic Solonchak we had not enough LF material in the Cz horizon to analyze the 14C activity. We agree with you that we should mention this in the figure caption and the Material & Methods section. This is going to be included in the revised MS.

SC #1:In figure 5 a little mistake has slipped in. The y-axisshould be PC2 instead of PC1. There we also wondered why the grey dots are notconsidered as they are quite

a lot.

A: Thank you for this correction. We are going to change that in the revised MS. The grey dots are important in the analysis. We have explained this in the answer to Referee #2. This is our response to Referee #2: "The PCA on neutral sugars was applied on the entire data set, i.e. neutral sugar data of all three soil types and all three fractions was analyzed in one PCA. This resulted in the biplot shown in Figure 5. To highlight differences between the soils we split the biplot into three panels and indicated the fractions of each soil by different colors. The biplot shows all considered data (i.e. the entire data set); this includes the grey dots which do not belong to the particular soil type of a panel. We decided to apply the PCA on the entire data set and not on the samples of each soil type separately, as the sample size would be too small to conduct a robust PCA for each soil type. This is a common approach and was applied in many previous studies."

SC #1:In the conclusion we would also appreciate an outlook for future studies. What would be important to look at?

A: An important issue would be to determine the water potential of all soils as the sum of matric potential + osmotic potential. Determination of the matric potential can be done by collecting undisturbed samples and measuring a soil retention curve. A time-series of soil moisture measurements could then be related to the soil water retention curve to obtain the matric potential at the particular soil moisture over the year. The osmotic potential can be determined via measurements of the electrical conductivity of the soil solution. By that we could verify whether the water stress, as indicated by a low water potential, is similar between the soils. Another promising approach would be to relate our results to measurements of enzyme activities. By that we would be able to directly determine whether the microbial activity is inhibited by salt stress or not. In combination with incubation studies of the bulk soil we could compare soil OM decomposition rates between the salt-affected and non-salt-affected soils. In the incubation studies we could adapt the soil moisture to the values observed in the field

to simulate field conditions. In the revised manuscript we are going to give a brief overview on that future research prospects.

SC #1: Some minor comments: - Strange starting sentence of the introduction "... soils...important...." !why do they get more important. They will get more frequent and-just to study them will get more important. Maybe "twice as" could be a nicer starting.

A: We agree with you and starting the sentence in the abstract with "Salt-affected soils will become more frequent in the next decades..." is a more precise statement. We are going to change that in the revised MS.

SC #1: Page 3/ line 42 !it is a german sentence; "Todate, these soils cover already an area..." do you need "already"?

A: "Already" indicates that the soils cover a considerable area worldwide.

SC #1: Page 6/ line 26 !units are at two lines

A: This manuscript is not yet text-edited. If published in Biogeosciences, text-editing will be done.

SC #1: Page 6/ line 33 !it is written Sect. 2.5, but chapters are notnumbered

A: The manuscript was written with a template offered by Copernicus Publications. This template does not include numbering of sections. But, if the manuscript gets published in Biogeosciences, numbering of sections will become necessary and thus we included the section number already.

SC #1: Page 9/ line 30 . . .very broad, peak broadening is related. . . ! you might make two sentences?

A: We agree with you and will correct this in the revised MS.

SC #1:Page 15/ line 19 This let's. . . ! informal english

A: In the revised MS we are going to change that to "This led us to the conclusion..."